# On the radiation belt location in the 23 – 24 solar cycles

**Alexei V. Dmitriev**[1,2]

[1]Institute of Space Science, National Central University, Jhongli, Taiwan,

[2]Skobeltsyn Institute of Nuclear Physics, Lomonosov Moscow State University, Moscow, Russia,

Corresponding author: Alexei Dmitriev (dalex@jupiter.ss.ncu.edu.tw)

**Abstract**

Within the last two solar cycles (from 2001 to 2018), the location of the outer radiation belt (ORB) was determined with using NOAA/Polar-orbiting Operational Environmental Satellite observations of energetic electrons with energies above 30 keV. It was found that the ORB was shifted a little (~1 degrees) in the European and North American sectors while in the Siberian sector, ORB was displaced equatorward by more than 3 degrees. The displacements corresponded qualitatively to the change of geomagnetic field predicted by the IGRF-12 model. However in the Siberian sector, the model has a tendency to underestimate the equatorward shift of ORB. The shift became prominent after 2012 that might be related to a geomagnetic jerk occurred in 2012 – 2013. The displacement of ORB to lower latitudes in the Siberian sector can contribute to an increase in the occurrence rate of mid-latitude auroras observed in the Eastern Hemisphere.

Keywords: electron radiation belt, secular geomagnetic variation, mid-latitude aurora

## 1. Introduction

The outer radiation belt (ORB) is populated by energetic and relativistic electrons trapped in the magnetosphere at drift shells above $L \sim 3$ (e.g. Ebihara and Miyoshi, 2011). The ORB is very dynamic and exhibits variations in a wide temporal range: short-term storm-time and local time variations, 27-day solar rotation, seasonal and solar cycle variations (e.g. Li et al., 2001; Baker and Kanekal, 2008; Miyoshi and Kataoka, 2011). During magnetic storms, the ORB is substantially disturbed and shifted earthward (Baker et al., 2016; Shen et al., 2017). The storm-time variation is the strongest one for both the ORB location and intensity (Baker and Kanekal, 2008). Magnetic storms produced by interplanetary coronal mass ejecta (ICME) and high-speed streams (HSS) of the solar wind from coronal holes. The seasonal variations with maxima at equinoxes can be explained by the effect of interplanetary magnetic field (IMF) orientation relative to the geomagnetic dipole (Li et al., 2001; O'Brien and McPherron, 2002; McPherron et al., 2009). ORB manifests prominent variations with the solar cycle (Fung et al., 2006; Baker and Kanekal, 2008). It was shown that the maximum of ORB is mostly distant from the Earth in solar minimum (Miyoshi et al., 2004) and it is closest to the Earth during solar maxima (Glauert et al., 2018).

Apparently, the intense variations mask relatively weak long-term changes related to a secular variation of the core and crustal magnetic fields. Recently, a number of authors reported significant changes in the Earth's magnetic field. The magnetic axial dipole has decreased over the past 175 years by 9% (e.g. Finlay et al., 2016). It was also shown that the north magnetic dip pole, the point where the magnetic field inclination is vertical, drifted from Canada toward Siberia with the speed rapidly increasing from 10 km/yr in 1990s to more than 50 km/yr at present (Chulliat et al., 2010; Thebault et al. 2015). From 1989 to 2002, most dramatic magnetic field changes of >50 nT/yr have been found in the Canadian Arctic and Eastern Siberia.

The effects of dipole decay and pole drift are predicted by International Geomagnetic Reference Field 12th generation (IGRF-12) model (e.g. Thebault et al. 2015). However in the Siberian sector, significant anomalies of the main geomagnetic field were found at high latitudes within the 80º-130º longitudinal range (Gvishiani et al., 2014). In this sense, independent verification of changes in the geomagnetic field at high and middle latitudes is required. Namely, the decrease of magnetic dipole

should result in a global equatorward shifting of the magnetospheric domains such as ORB and auroral region. The drift of the north magnetic pole should cause a decrease(increase) of ORB and auroral latitudes in the Siberian(North American) sectors.

The long-term changes in the location of auroral region were reported by Smith et al. (2017). They analyzed the latitudinal location of auroral electro jets (AEJs) and revealed a prominent latitudinal displacement of the AEJs by several degrees in the years 2004 – 2014 relative to the previous solar maxima in 1970 and 1980. Namely, in the Siberian sector, AEJ shifted to lower latitudes and in the American sector, AEJ shifted to higher latitudes. The opposite shifts in different sectors cannot be explained by the solar cycle variation and, thus, it has been attributed to the core and crustal magnetic fields. On the other hand, the technique of auroral precipitations is hard to use for tracing of the long-term geomagnetic variations because of high variability in the intensity, location and extension of aurora (e.g. Cresswell-Moorcock et al., 2013; Smith et al.; 2017).

An additional support of prominent changes in the geomagnetic field can be found from a sudden increase of occurrence of aurora borealis during the years of 2015 to 2017. There were numerous reports about aurora borealis observed at middle latitudes in the North America, Europe and Russia. Table 1 lists the days when discrete aurora was detected in big Russian cities Moscow (geographic location 55°45N 37°37E), St. Petersburg (geographic location 59°57N 30°18E) and Novosibirsk (geographic location 55°01N 82°55E). It is important to note that while in the North American region, the mid-latitude discrete aurora is observed quite often, this phenomenon is rare at lower magnetic latitudes such as in the regions of Central Europe and in particular in Central Russia (MacDonald et al., 2015; Vázquez et al., 2016). The previous low-latitude aurora borealis was observed during extremely strong geomagnetic storms with minimum $Dst$ < -300 nT on October - November 2003 (e.g. Shiokawa et al., 2005; Mikhalev et al., 2004).

In contrast, magnetic storms in 2015 – 2017 were not very intense, as one can see in Table 1. The strongest storm on 17 – 18 March 2015, so-called St. Patrick's Day storm, had minimum $Dst$ of -220 nT (e.g. Kataoka et al., 2015). During the St. Patrick's Day storm, aurora borealis was observed worldwide in North America, Central Europe (e.g. "Strongest geomagnetic storm of SC24 sparks spectacular aurora display" at https://watchers.news/2015/03/18/) and in a number of

cities in Central Russia and Siberia (e.g.
https://www.rt.com/news/241845-aurora-borealis-central-russia/). Case et al. (2015) found that
during the storm, the discrete aurora was observed at unusually low latitudes, which were much
lower than those predicted by models of Roble and Ridley (1987) and Newell et al. (2010).
The aurora is produced by charged particles precipitating from the magnetosphere to the
high-latitude atmosphere. The charged particles move along the magnetic field lines and, thus, the
location of precipitation is controlled both by the location of source and by the geomagnetic field
configuration. In the present study, we analyze the configuration of the magnetosphere by using
observations of energetic electrons from ORB. At low heights, the ORB electrons are observed at
middle to high latitudes adjacent to the region of auroral precipitations (Lam et al., 2010). Here we
use experimental data on energetic electrons measured by several low-earth orbit (LEO) polar
orbiting satellites during the time period from 2001 to 2016. The method of analysis is described in
section 2. The results are presented and discussed in sections 3 and 4, respectively. Section 5 is
conclusions.

**2. Method**
Energetic electrons in energy ranges >30 keV, >100 keV and >300 keV are measured at LEO by
the Medium Energy Proton and Electron Detector (MEPED) instruments on board the
NOAA/Polar-orbiting Operational Environmental Satellite (POES) satellites (Evans and Greer,
2004; Asikainen and Mursula 2013). Six POES satellites NOAA-16, NOAA-17, NOAA-18,
NOAA-19, METOP-01 and METOP-02 (hereafter, P6, P7, P8, P9, P1 and P2, respectively) have
Sun-synchronous orbits at altitudes of ~800-850 km in different local time sectors. Different POES
satellites were operating during different years as shown in Table 2.
The outer magnetosphere and ORB are very dynamic regions, which are directly controlled by
highly variable solar wind plasma streams and interplanetary magnetic field (IMF). As a result, the
location of ORB and its high-latitude projection to the heights of LEO vary substantially (e.g.
Dmitriev et al., 2010; Rodger et al., 2010). Namely, a strong local time variation is related to the
global day-night asymmetry of the magnetosphere such that ORB is observed at higher latitudes
during daytime. Variation of geomagnetic tilt angle also causes a change of the ORB latitudinal
location. Interplanetary and geomagnetic disturbances result in a prominent equatorward shift of
ORB.
In order to eliminate the disturbing factors, we consider so-called quiet days. Figure 1
demonstrates an example of geomagnetic conditions and measurements of the solar wind plasma
and IMF acquired from Wind upstream monitor during quiet day on 23 June 2006. At that day, the
solar wind velocity was slow (~310 km/s), solar wind dynamic pressure was slightly varying about
~1.6 nPa, IMF had northward orientation that resulted in very quiet geomagnetic activity (*AE* <
100 nT, *Dst* ~ 0 nT).
The list of quiet days selected in the time interval from 2001 to 2018 is presented in Table 2. The
solar wind data were acquired from Wind upstream monitor. The selection of quiet days was based
on the following criteria:
1. The *Dst* variation was close to 0 and *AE* index was smaller than 200 nT, i.e. the geomagnetic
activity was very weak.
2. The solar wind dynamic pressure *P*d varied slightly around its average values falling in the
range from ~1 to 2 nPa.
3. The solar wind speed was <400 km/s and the amplitudes of negative IMF Bz were weak (<4 nT).
Note that the solar wind with the speed of V > 400 km/s is often associated with HSSs from
coronal holes. Fast solar wind streams initiate the Kelvin-Helmholtz instability at the
magnetopause and also produce recurrent magnetic storms, which are accompanied by
intensification of wave activity in the outer magnetosphere that results in effective acceleration and
radial transport of the ORB electrons (Engebretsone et al., 1998; Tsurutani et al., 2006; Horne et al.,
2007; Su et al., 2015).
4. The quiet days were chosen as long as possible after magnetic storms such that storm-time
disturbances of ORB had time to relax. Usually, the quiet days occurred after long-lasting recovery
phase of recurrent magnetic storms (Suvorova et al., 2013).
The local time variation of ORB latitudinal location was minimized by a choice of narrow LT
sector around noon (from 10 to 14 LT). We chose quiet days around June solstice in order to
minimize the tilt angle variations. Note that June of 2003 and 2007 was very disturbed and there
were no quiet days selected for those years.
Figure 2 shows an example of NOAA/POES measurements of energetic electrons in geographic
coordinates during the quiet days on 23 June 2006 and 3 June 2016. The geographic maps are
composed from data retrieved over multiple orbits of the NOAA/POES satellites in the noon sector
(12±2 LT). For each bin of 3° in longitudes and 0.5° in latitudes, we calculate the average flux of
electrons measured by the 90° detector of the MEPED instrument. At high latitudes, the detector
observes trapped electrons with pitch angles close to 90°, i.e. near the mirror points.
The limitation of ORB measurements at given local time is originated from fixed local time of
POES satellites at sun synchronous orbits. As one can see in Figure 2 and Table 2, large statistics
in the Northern hemisphere can be obtained from a number of POES satellites moving in 2-hour
vicinity of local noon around the June solstice. ORB can be easily identified as a wide belt of
intense electron fluxes at high litutudes. At middle latitudes, in longitudinal ranges from ~90°E to
180°E in the Eastern Hemisphere and from ~80°W to 180°W in the Western Hemisphere, one can
also see intense electron fluxes from the inner electron belt and a slot region between the outer and
inner belts. The slot region is almost vanished in the maps of subrelativistic electrons with energies
>300 keV. Qualitative examination of the ORB location in Figure 2 reveals that in the Eastern
Hemisphere, the outer electron belt in 2016 is located few degrees lower in latitudes than that in
the year 2006. Most obvious difference can be found for the slot region, which corresponds to the
low-latitude boundary of ORB.
For quantitative determination of the ORB latitudinal displacement, we analyze electron fluxes in
4° vicinities of three longitudes: 80°W (American sector), 0°E (European sector) and 100°E
(Siberian sector). Figure 3 shows latitudinal profiles of >30 keV; >100 keV and >300 keV electron
fluxes with pitch angles of ~90° observed by the NOAA/POES satellites around given longitudes
during the quiet days in the years from 2001 to 2018. One can easily identify the maximum of
ORB at high latitudes and the slot region at middle latitudes for the American and Siberian sectors.
Above the Europe, the slot region is not detected at altitudes of the NOAA/POES orbit.
It should be noted that after the year 2014, the experimental data on electrons detected by POES is
presented in a different format such that the energy channels of electrons are different from those
presented earlier: >40 keV instead of >30 keV, >130 keV instead of >100 keV, and >290 keV
instead of >300 keV. Because of that cross-calibration of the electron detectors is difficult. On the
other hand, the difference in energies is not very large and, thus, it should not affect strongly the
location of ORB. At least the differences are much smaller than the steps between the channels.
Therefore, the complex analysis of all three electron channels allows minimization of this effect.

**3. Results**
In Figure 3, the ORB maxima in the American, European and Siberian sectors can be found in the
ranges of latitudes from 50° to 58°, from 64° to 70° and from 62° to 74°, respectively. We
determine geographic latitude of the maxima for each year with the accuracy of 0.5° to 1°. One can
see that the location as well as the intensity of the maximum varies from year to year. The intensity
is minimal during the solar minimum in 2009. The fluxes of >300 keV electrons (Figure 3c) were
very weak such as determination of the ORB was very difficult. In addition, the ORB maximum
above Siberia could not be determined in 2011 because of limited statistics.
Qualitatively, the position of ORB maximum above Siberia is more close to 70° and 65°,
respectively, in 2001 - 2010 and in 2012 – 2018. Above the Europe and North America, variation
of the ORB location is more random. The fluxes of >30 keV electrons in the outer region of ORB
are very dynamic because of strong contribution from the auroral population. The latter produced
additional maxima at latitudes above 70° and 55°, respectively, in the European-Siberian and
American sectors. The additional maxima were very intense in the years 2008, 2010 and 2017 that
made difficult to determine the actual location of the ORB. In those cases, we chose the maximum
located at lower latitude. This choice gives a good agreement with the ORB maximum location for
the >100 keV electrons and especially subrelativistic >300 keV electrons, which are practically
free from the auroral contamination.
In Figure 3, one can clearly see the slot region between the outer and inner electron belts in the
latitudinal ranges 45° - 50° and 45° - 50° above North America and Siberia, respectively. This
structure can be well identified and numerically determined, excepting >300 keV electrons. In the
case of slot region, the low-latitude edge of ORB is determined as the first high-latitude point of
gradual flux enhancement after the slot minimum. Apparently, the electron flux enhancements peak
in the maximum of ORB, which location can be determined unambiguously. In the European
sector and for the electrons with energies >300 keV, the criterion for determination of the inner
edge is not so obvious. It is difficult to define a threshold flux because of strong solar cycle
variations of electron fluxes. In this case, the inner edge can be determined as the lowest latitude of
gradual decrease of electron fluxes from the ORB maximum toward lower latitudes. As one can
see in Figure 3, the inner edge separates usually the background noise with sharply varying fluxes
at lower latitudes from smooth and fast increase of ORB fluxes at higher latitudes. Geographic
latitude of the inner edge is determined for each year with the accuracy varying from 0.5° to 1°. In
the American sector, the inner edge of ORB is situated at lowest latitudes from 43° to 51°, in the
European sector – from 55° to 63°, and in the Siberian sector – at highest latitudes from 58° to 65°.
In Figure 3, one can find that the latitude of ORB edge above Siberia decreases with years from
~65° to 60° for all energy range of electrons. The change of ORB location above the Europe and
North America is not so obvious.
Figure 4 and Figure 5 show long-term variations in the location of ORB and corresponding
predictions of the IGRF-12 model during 17 years from 2001 to 2018. The prediction of IGRF-12
model was calculated in the following manner. First, we took a point with given geographic
coordinates and calculated its magnetic coordinates for the quiet day on 29 June 2001 using the
IGRF model of epoch 2000. Namely, for the ORB maximum, we took points (70°N, 80°W), (66°N,
0°E) and (54°N, 100°E), respectively, for the American, European and Siberian sectors and
calculated their geomagnetic coordinates (64.12°N, 11.44°W), (67.05°N, 95.66°E) and (59.5°N,
174.3°E), respectively. For the inner edge of ORB, we took, respectively, (46.5°N, 80°W), (59°N,
0°E) and (63°N, 100°E), with corresponding geomagnetic coordinates (56.62°N, 10.61°W)
(60.59°N, 89.34°E) and (52.47°N, 173.7°E). Then we supposed that the geomagnetic coordinates
of the points do not change with time and we used them to calculate geographic coordinates from
the IGRF-12 model for corresponding quiet days listed in Table 2. The geographic coordinates of a
point with given magnetic coordinates should be changed with time because of long-term variation
of the geomagnetic field.
In Figure 4 and Figure 5, one can see that the ORB maximum and inner edge of >30 keV electrons
are usually located at higher latitudes than those of >100 keV electrons, and the ORB of
subrelativistic >300 keV electrons is located at lowest latitudes. Note that the location of ORB
maximum for >30 keV electrons is scattered significantly and it is different from those for the
more energetic electrons because of substantial contamination from the auroral electrons. In
contrast, the ORB maxima and inner edge of >100 keV and >300 keV electrons demonstrate very
similar dynamics. The location of ORB manifests the well-known solar cycle variation: the
latitudes of ORB maximum and inner edge have a tendency to be highest around solar minimum in
2008 – 2009 and lowest during solar maxima in the years 2001 and 2012 – 2013. Note that the
maximum phases of the 23$^{rd}$ and 24$^{th}$ solar cycles occurred in the years 2000 - 2001 and in 2012 –
April 2014, respectively. The years 2008 – 2009 are the solar minimum phase. The declining
phases lasted from 2003 to 2007 and from 2014 to 2018. In Figures 4 and 5, one can see that
during the declining phase of the current 24$^{th}$ solar cycle (especially in the years 2016 – 2018), the
behavior of the ORB maximum and inner edge is different from that during the declining phase of
the previous 23$^{rd}$ solar cycle. Namely, their latitudes increased only slightly or even decreased
above North America and especially above Siberia.
Unfortunately, there is no any model of the ORB location variation with the solar cycle because the
driving mechanisms are not well established. On the other hand, the long-term variation in
IGRF-12 is almost linear function of the year, as one can see in Figures 4 and 5. Hence, as a first
approach for comparative analysis, the variations of ORB location with years are considered as
random around a linear function (indicated by dashed strait lines in Figures 4 and 5):
$$\lambda = a * \text{year} + b, \text{ (1)}$$
where $\lambda$ is the latitude of maximum or inner edge of ORB. The slope $a$, parameter $b$ and their
standard errors are calculated from a linear regression for various longitudinal regions and various
energies of electrons. The results are presented in Tables 3 and 4 for the ORB maximum and the
inner edge, respectively. The linear fits are compared with geomagnetic field trends predicted by
the IGRF model. The trends are also fitted by a linear function with the slope $a_{IGRF}$.
In the American sector (see Figure 4a), the latitude of ORB maximum demonstrates a little
decrease of about 1° while the IGRF-12 model predicts an increase of ~1°. The decrease results
from relatively low latitudes, where the ORB maximum is located from 2013 to 2018. The location
of inner edge of ORB in the American sector (see Figure 5a) does not practically change within the
experimental uncertainty of ~1°. Note that in both cases, the slope $a$ has very large errors (see
Tables 3 and 4) such that the slope of IGRF trend $a_{IGRF} = 0.06$ falls almost into the error ranges.
Hence, from the statistical consideration one can conclude that the model prediction does not
contradict to the observations.
In the European sector (Figures 4b and 5b), the IGRF-12 model predicts very small change of 0.3°
in the ORB location with the slope $a_{IGRF} \sim 0.02$ that is in good agreement with the ORB maximum
dynamics. The location of ORB inner edge for electrons with energies >30 keV and >100 keV
demonstrates an increase of ~3°. However, the slope of increase is determined with a substantial
error of up to 50% (see Table 4) that produces an increase by only ~1.5°. In addition, the >300 keV
electrons follow the model and do not exhibit any prominent trend. Hence in the European sector,
the IGRF model predicts the ORB dynamics with sufficient accuracy.
In the Siberian sector, the IGRF model predicts ~1° decrease in the latitude of ORB maximum and
inner edge (see Figures 4c and 5c) with the slope $a_{IGRF} \sim -0.06$. From the POES observations, we
find that the ORB maximum is displaced to lower latitudes by at least ~3° in all electron energy
channels: from ~69° to ~66° for >300 keV electrons, from ~70° to 66° for >100 keV electrons and
from ~71° to 67° for >30 keV electrons (see Figure 4c). The difference is related to very low
latitudes (~67° and less) of the ORB maximum during solar maximum and on the declining phase
of the current 24th solar cycle in the years 2012 - 2013 and 2016 - 2018, respectively. In the solar
maximum and on the declining phase of the previous 23rd solar cycle (the years 2001 and 2004 -
2006), the ORB maximum was located at higher latitudes (above 67°). In Table 3, the slopes for all
energy ranges are steeper than the slope of IGRF. Note that the errors in determination of the slope
$a$ are ~50%. Hence statistically, the decrease of latitude might be two times smaller, i.e. ~1.5° to 2°.
This decrease is slightly larger than 1° of the model prediction, within 0.5° to 1° statistical
uncertainty in determination of latitude.
Similar pattern can be found for the inner edge of ORB in the Siberian sector (see Figure 5c).
Namely, the IGRF model predicts a decrease of ~1° with the slope $a_{IGRF}$ ~ -0.06. The inner edge
was shifted toward lower latitudes by ~3°, ~2° and ~1°, respectively, for >30 keV, >100 keV and
>300 keV electrons. From Table 4, one can see that the slopes $a$ are steeper than $a_{IGRF}$. The slopes
are calculated with errors of ~30% and ~20%, respectively, for >30 keV and >100 keV electrons. It
means that the decrease in latitude might be ~2° (instead of ~3°) and ~1.5° (instead of ~2°),
respectively. These values are again larger than 1° of the model prediction. Hence, there is a
tendency that the change in the latitudinal location of ORB maximum is underestimated by the
model. This fact indicates that during 17 years from 2001 to 2018, ORB is abnormally displaced
toward the lower latitudes in the Siberian sector.
It is interesting to point out the year 2017, when the maximum and inner edge of ORB shifted to
very low latitudes of 62° and ~59° respectively. The shift was observed during two quiet days on 9
and 10 June 2017. Similar pattern of displacement can be found on the declining phase of the
previous 23rd solar cycle in the year 2005, when the ORB suddenly shifted equatorward by more
than ~2°. Note that if we exclude the year 2017 from the linear fitting then the results are not
practically changed because ORB is located at relatively low latitudes during the years 2012 to

2018.


**4. Discussion**
We have found up to 4° equatorward displacement of the ORB in the Siberian sector. The
displacement is larger than that predicted by the IGRF-12 model. The difference is statistically
significant. It might result both from a change of the geomagnetic field and from changes of
driving parameters such as geomagnetic activity, the tilt angle, IMF $Bz$ and solar wind dynamic
pressure. It is well known that those parameters affect the latitudinal location of domains in the
magnetosphere. The effect of geomagnetic activity was eliminated by the choice of quiet days. The
other drivers are considered below.
The tilt angle in the noon region at given longitude (80°W, 0°E and 100°E) varies a little (<2°)
during the June month. The change of local time in 2-hour vicinity of noon produces ~5° variation
of the tilt angle. The tilt angle variations of a few degrees result in a tiny change of ~0.1° in the
ORB latitude (e.g. Dmitriev et al., 2010). Hence, we can neglect the effect of tilt angle.
The effect of solar wind parameters, including IMF $Bz$ and dynamic pressure ($P$d), to the ORB
location is not obvious. It is found that the slot region location can be related to the plasmapause
but the relation is ambiguous (Darrouzet et al., 2013; Baker et al., 2014). We can make indirect
estimation of the effect using a dependence of the cusp location from the solar wind parameters
(Kuznetsov et al., 1993; Newell et al., 2006). The equatorward edge of the cusp separates the open
and close magnetic filed lines in the dayside magnetosphere. Hence the latitude of the equatorward
edge can be considered as a proxy of the ORB outer edge. In the first approach, we assume that the
effect of solar wind parameters to the ORB location can be represented by the dynamics of the
ORB outer edge or the cusp equatorward edge. It can be shown that $Bz = $ -4 nT results in less than
0.5° equatorward shift of the cusp and a change of $P$d from 1 to 2 nPa results in ~0.2° decrease in
the latitude of the cusp equatorward edge. Hence, the effects of both $P$d and IMF Bz are several
times weaker than the difference of 3°.
Another possible effect is the solar cycle variation. Variations of the ORB location from cycle to
cycle and during different phases of solar cycles are still poorly investigated. It was well
established that during solar minima and maxima, the ORB is located, respectively, at highest and
lowest latitudes (Miyoshi et al., 2004; Glauert et al., 2018). From these findings, we can speculate
that lower(higher) solar activity results in an increase (a decrease) of the ORB latitudes. In Figure
3, one can see that the intensities of electrons are weaker after the beginning of the 24th solar
maximum in 2012 in comparison with the 23rd solar cycle. Note that the 23rd solar cycle was
stronger than the 24th one. Following this logic, the ORB should be located at relatively higher
latitudes during the weak 24th solar cycle than during the strong 23rd solar cycle.
In Figure 6, the outer radiation belt location is compared during the maximum and declining
phases of the solar cycles 23rd (the years 2001 – 2006) and 24th (the years 2013 – 2018). During
those time intervals, the sunspot numbers for the both cycles correlate very well. The ORB
location demonstrates also very similar solar cycle variations. The ORB latitude increased after the
solar maximum in 2001 – 2002 (and in corresponding years 2013 – 2014). During those years, the
ORB location was quite close for the both cycles. The difference of ~1° can be explained by the
secular variation predicted by the IGRF model. During the declining phase in 2004 – 2005 (2015 –
2017), the ORB was shifted to lower latitudes and then it moved slightly poleward in 2006 (2018),
when the solar minimum was approached.
From Figure 6, one can clearly see that on the declining phase of the 24[th] solar cycle, the outer
radiation belt is located at latitudes lower by several decrees than those during the 23[rd] solar cycle.
It is interesting to point out the year 2017, when the maximum and inner edge of ORB were shifted
to very low latitudes of 62° and ~59° respectively. The shift was observed during two quiet days on
9 and 10 June 2017. Similar pattern of strong displacement by more than ~2° can be found on the
declining phase of the previous 23rd solar cycle in 2005, the year corresponding to the similar
stage of solar activity. Hence, the ORB dynamics in the year of 2017, as well as during the whole
declining phase from 2014 to 2018, was not anomalous in the sense of solar cycle variations.
However, the ORB latitudes were abnormally low. The difference of several degrees cannot be
explained by the IGRF model. As a result, we have found totally opposite effect: ORB over Siberia
is located at lower latitudes during the weak 24th solar cycle than during the strong 23rd solar
cycle. It should be noted that if one excludes the year 2017 from the linear fitting then the results
are not practically changed because ORB is located at relatively low latitudes during practically
whole declining phase of the 24th solar cycle.
From the above, we can conclude that the difference between the observations and predictions can
be rather originated from anomalous dynamics of the geomagnetic field. This idea is supported by
the observations of ORB location over the Europe and North America, where the ORB
displacement is well predicted by the IGRF-12 model. An additional support can be found from
results of long-term magnetic observations in Siberia where significant anomalies of the main
geomagnetic field have been revealed in the 80°-130° longitudinal range (Gvishiani et al., 2014).
Namely, the IGRF-12 model predicted the magnetic field up to 300 nT stronger than that measured
by ground based magnetic stations that was close to 0.5% of the total magnetic filed in this region.
For the geodipole, stronger magnetic field corresponds to higher latitudes.
In Figures 4c and 5c, one can see that the decrease of ORB latitude in the Siberian sector is most
prominent after 2012. On the other hand in the years 2012 –2013, a sudden change was found in
the acceleration of secular variation in the geomagnetic field (Finlay et al., 2015). Analyzing time
interval from 1999 to 2015, Finlay et al. (2015) revealed 3 pulses in time evolution of the mean
square secular acceleration power: in 2006, in 2009 and in 2012 – 2013. Chulliat et al. (2015)
attribute these pulses, or so-called sharp geomagnetic "jerks", to magnetic field variations
originating in the Earth's core. We can assume that the abnormal ORB displacement might be
related to the geomagnetic jerks. We can assume that the abnormal ORB displacement might be
related to the geomagnetic jerks. Though, there is no prominent change in the ORB location in 2006, one
can indicate very high latitude of ORB in 2009. Note that the jerk in 2009 coincided with the abnormally
deep solar minimum and, hence, it could be hard to distinguish between the two effects. On the other hand,
we have found significant change in the ORB dynamics after 2012 – 2013.
The several degrees equatorward displacement of ORB in the Siberian sector indicates an
equatorward shifting of all domains in the magnetosphere, including the region of auroral
precipitations. Apparently, the shifting contributes to the increase in occurrence rate of the
mid-latitude auroras in Siberia and, perhaps, in entire Russia. In addition, Finlay et al. (2015)
expect that the next jerk might occur around 2016. We do not have any reports about the recent
jerks yet. But very strong decrease of the ORB latitude observed in 2017 might indicate the sudden
change in the geomagnetic field.

**5. Conclusions**
NOAA/POES observations of electrons with energies of few tens and hundreds of keV allowed
revealing and measure a latitudinal displacement of the outer radiation belt during last 18 years.
The displacement corresponds qualitatively to the change of geomagnetic field predicted by the
IGRF-12 model. However in the Siberian sector, the model has a tendency to underestimate the
equatorward shift of ORB. The shift became prominent after 2012 that might be related to the
geomagnetic jerk occurred in 2012 – 2013. The increase in the occurrence rate of mid-latitude
auroras in the Eastern Hemisphere can be explained, at least partially, by the equatorward
displacement of the high-latitude projection of the magnetosphere domains.

**Acknowledgments** The authors thank a team of NOAA's Polar Orbiting Environmental Satellites
for providing experimental data about energetic particles, the CDAWEB for providing the Wind
solar wind data, Kyoto World Data Center for Geomagnetism
(http://wdc.kugi.kyoto-u.ac.jp/igrf/point/index.html) for providing the geomagnetic indices and
computation of the IGRF12 model, and WDC-SILSO, Royal Observatory of Belgium, Brussels for
providing sunspot numbers (http://www.sidc.be/silso/datafiles). The work was supported by grant
MOST-106-2111-M-008-015-, R&D foundation from National Central University and partially by
grant NSC103-2923-M-006-002-MY3/14-05-92002HHC_a of Taiwan - Russia Research
Cooperation.

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

**Table 1.** Observations of discrete aurora in Russia in the years 2015 to 2016

| Date | min Dst, nT | City | Geomagnetic location | Reference |
|---|---|---|---|---|
| 2015 March 17-18 | -220 | Moscow | 51°16N 122°06E | Ref1 |
| 2015 June 22-23 | -200 | Moscow | 51°16N 122°06E | Ref2 |
| 2015 August 16-17 | -84 | St. Petersburg | 56°23N 117°36E | Ref3 |
| 2015 October 7-8 | -120 | St. Petersburg | 56°23N 117°36E | Ref4 |
| 2016 February 17-18 | -50 | St. Petersburg | 56°24N 117°37E | Ref5 |
| 2016 April 3-4 | -50 | St. Petersburg | 56°24N 117°37E | Ref6 |
| 2016 August 24-25 | -80 | St. Petersburg | 56°24N 117°37E | Ref7 |
| 2017 September 7-8 | -124 | Novosibirsk | 45°56N 160°07E | Ref8 |
| 2017 November 7-8 | -74 | St. Petersburg | 56°25N 117°38E | Ref9 |

Ref1 - www.dp.ru/a/2015/03/18/Severnoe_sijanie_uvideli_zh/
Ref2 - www.dp.ru/a/2015/06/23/Severnoe_sijanie_uvideli_v/
Ref3 - http://47news.ru/articles/92419/
Ref4 - www.dp.ru/a/2015/10/08/Severnoe_sijanie_v_Peterbu/
Ref5 - www.fontanka.ru/2016/02/17/058/
Ref6 - www.dp.ru/a/2016/04/03/ZHiteli_Peterburga_deljatsja/
Ref7 - www.fontanka.ru/2016/08/24/035/ and www.topnews.ru/news_id_92986.html
Ref8 - http://www.ntv.ru/video/1515160/
Ref9 - https://www.fontanka.ru/2017/11/07/134/


**Table 2.** List of quiet days in June selected for POES observations of the outer radiation belt.

| Year | Day in June | Start UT | Duration, hours | V* km/s | Pd** nPa | $Bz_{min}$ nT | POES Satellites[#] |
|---|---|---|---|---|---|---|---|
| 2001 | 29 | 0 | 24 | 350 | 1.6 (1.0 – 3.2) | 0.6 (-4) | P6 |
| 2002 | 28 | 0 | 24 | 340 | 1.2 (0.8 – 1.8) | 2.2 (-3) | P6 |
| 2004 | 24 | 12 | 24 | 330 | 1.1 (0.5 – 2.5) | 1.2 (-2) | P6, P7 |
| 2005 | 21 | 0 | 18 | 350 | 0.9 (0.5 – 2.0) | 3.1 (-4) | P6, P7, P8 |
| 2006 | 23 | 0 | 24 | 310 | 1.6 (1.1 – 2.3) | 3.4 (-1) | P6, P7, P8 |
| 2008 | 13 | 0 | 24 | 310 | 1.5 (0.8 – 1.9) | 1.8 (-0.8) | P2, P7, P8 |
| 2009 | 17 | 0 | 24 | 300 | 1.1 (0.5 – 1.7) | 1.9 (-3) | P2, P7, P8, P9 |
| 2010 | 12 | 0 | 24 | 350 | 1.1 (0.6 – 2.4) | 0.2 (-2) | P2, P7, P8, P9 |
| 2011 | 28 | 6 | 24 | 390 | 0.8 (0.5 – 1.7) | 1.8 (-2) | P2, P6, P8, P9 |
| 2012 | 15 | 0 | 24 | 320 | 0.8 (0.5 – 1.3) | 0.0 (-3) | P2, P6, P8, P9 |
| 2013 | 16 | 0 | 24 | 330 | 0.9 (0.6 - 1.5) | 1.0 (-3) | P2, P6, P8, P9 |
| 2014 | 1 | 0 | 36 | 300 | 1.7 (1.1 – 4.0) | 1.5 (-4) | P1, P2, P9 |
| 2015 | 4 | 0 | 24 | 280 | 1.0 (0.7 – 1.7) | 0.9 (-3) | P1, P2, P9 |
| 2016 | 3 | 0 | 24 | 300 | 1.0 (0.7 – 1.4) | -0.3 (-3) | P1, P2, P9 |
| 2017 | 9 | 6 | 24 | 310 | 1.9 (1.0 – 2.6) | -1.3 (-4) | P1, P2, P9 |
| 2018 | 12 | 8 | 24 | 300 | 1.3 (0.9 – 2.0) | 0.0 (-4) | P1, P2, P9 |

*Daily average of the solar wind velocity
**Daily average of the solar wind dynamic pressure and its minimum and maximum in brackets
$Daily average Bz component of the interplanetary magnetic field and Bz minimum in brackets
#POES satellites observed the outer radiation belt

**Table 3.** Coefficients of the best linear fit of the latitudinal change of the ORB maximum location with years for various longitudes and energy of electrons

| Longitude, deg | Energy, keV | $a_{IGRF}$, deg/year | $a$, deg/year |
|---|---|---|---|
| -80 | >30 | 0.06 ± 0.003 | -0.153 ± 0.112 |
| -80 | >100 | 0.06 ± 0.003 | -0.069 ± 0.097 |
| -80 | >300 | 0.06 ± 0.003 | -0.057 ± 0.084 |
| 0 | >30 | 0.018 ± 0.001 | 0.021 ± 0.089 |
| 0 | >100 | 0.018 ± 0.001 | -0.032 ± 0.063 |
| 0 | >300 | 0.018 ± 0.001 | -0.027 ± 0.042 |
| 100 | >30 | -0.06 ± 0.003 | -0.265 ± 0.119 |
| 100 | >100 | -0.06 ± 0.003 | -0.208 ± 0.106 |
| 100 | >300 | -0.06 ± 0.003 | -0.167 ± 0.084 |

**Table 4.** Coefficients of the best linear fit of the latitudinal change of the ORB inner edge location with years for various longitudes and energy of electrons.

| Longitude, deg | Energy, keV | $a_{IGRF}$, deg/year | $a$, deg/year |
|---|---|---|---|
| -80 | >30 | 0.06 ± 0.003 | -0.029 ± 0.065 |
| -80 | >100 | 0.06 ± 0.003 | -0.021 ± 0.059 |
| -80 | >300 | 0.06 ± 0.003 | -0.014 ± 0.063 |
| 0 | >30 | 0.019 ± 0.001 | 0.195 ± 0.107 |
| 0 | >100 | 0.019 ± 0.001 | 0.241 ± 0.078 |
| 0 | >300 | 0.019 ± 0.001 | 0.032 ± 0.069 |
| 100 | >30 | -0.06 ± 0.003 | -0.183 ± 0.058 |
| 100 | >100 | -0.06 ± 0.003 | -0.211 ± 0.037 |
| 100 | >300 | -0.06 ± 0.003 | -0.097 ± 0.069 |

     **Figure captions**

Figure 1. Solar wind and geomagnetic conditions on 22 to 24 June 2006 (from top to bottom):
solar wind bulk velocity V; solar wind dynamic pressure Pd; interplanetary magnetic field
magnitude B (blue dotted curve) and Bz component (black solid curve); auroral electrojet index
AE; storm-time *Dst* index. The day on 23 June (indicated by vertical red dashed lines) is very quite
in the solar wind and geomagnetic parameters.

Figure 2. Geographic maps of energetic electron fluxes with energies >300 keV (a,b), >100 keV
(c,d), >30 keV (e,f) and pitch angles of ~90° observed by POES satellites at height of ~850 km in 2
hour vicinity of local noon (left column) on 23 June 2006 and (right column) on 2 June 2016. The
solid wide curve indicates the geomagnetic equator. The outer and inner electron belts and a slot
region between them are clearly seen (excepting of >300 keV electrons), respectively, at high and
middle latitudes in the longitudinal range from ~90° E to ~80°W.

Figure 3. Latitudinal profiles of electron fluxes with pitch angles of ~90° observed by POES
satellites during quiet days in different years at height of ~850 km in vicinity of local noon at
longitudes around 100°E (red circles), 0°E (blue crosses), and 80°W (black diamonds) for various
energy channels: (a) >30 keV, (b) >100 keV, and (c) >300 keV. Vertical dashed and solid lines
indicate latitudes of the maximum and inner edge of the outer radiation belt, respectively.

Figure 4. Geographic latitude of the maximum of the outer radiation belt measured at height of
~850 km during geomagnetic quiet days around 80°W (a), 0°E (b), and 100°E (c) for electrons
with energies of >30 keV (red circles), >100 keV (blue crosses), and >300 keV (green triangles).
Dashed curves of corresponding colors show the best linear fit of the latitudinal change of the
maximum location with years (see Table 3). Solid black curves show the latitudinal change
predicted by the IGRF model of corresponding epochs (see details in the text). The grey curve
shows sunspot number (right axis).

Figure 5. The same as Figure 4 but for the inner edge of the outer radiation belt. Coefficients of the
best linear fit are presented in Table 4.

Figure 6. Geographic latitude of the inner edge (a) and maximum (b) of the outer radiation belt
measured during geomagnetic quiet days at height of ~850 km around longitude of 100°E for
electrons with energies of >30 keV (circles), >100 keV (crosses), and >300 keV (triangles).
Bottom panels show the sunspot number in the 23[rd] solar cycle (the years 2001 – 2006, black
curves) and in the 24[th] solar cycle (the years 2013 – 2018, blue curves). The outer radiation belt
location is shown by black and red symbols, respectively, for the 23[rd] and 24[th] solar cycles. It can
be seen that on the declining phase of the 24[th] solar cycle, the outer radiation belt is systematically
located at lower latitudes than that during the 23[rd] solar cycle.


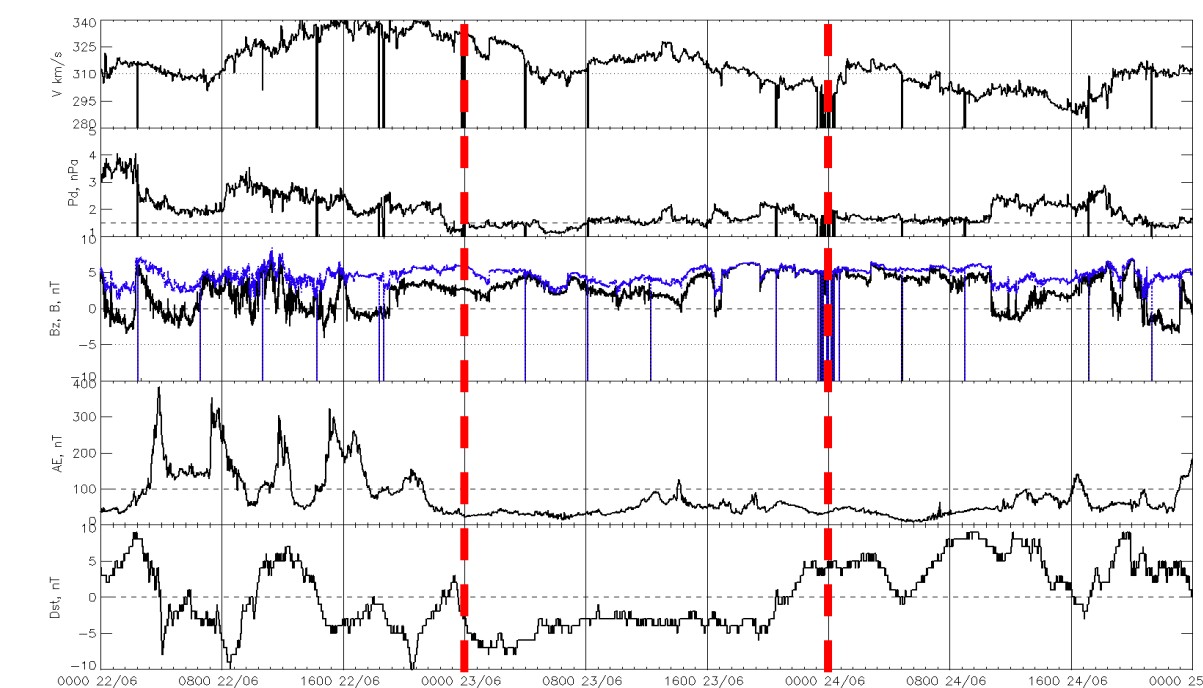

Figure 1. Solar wind and geomagnetic conditions on 22 to 24 June 2006 (from top to bottom): solar wind bulk velocity V; solar wind dynamic pressure Pd; interplanetary magnetic field magnitude B (blue dotted curve) and Bz component (black solid curve); auroral electrojet index AE; storm-time *Dst* index. The day on 23 June (indicated by vertical red dashed lines) is very quite in the solar wind and geomagnetic parameters.

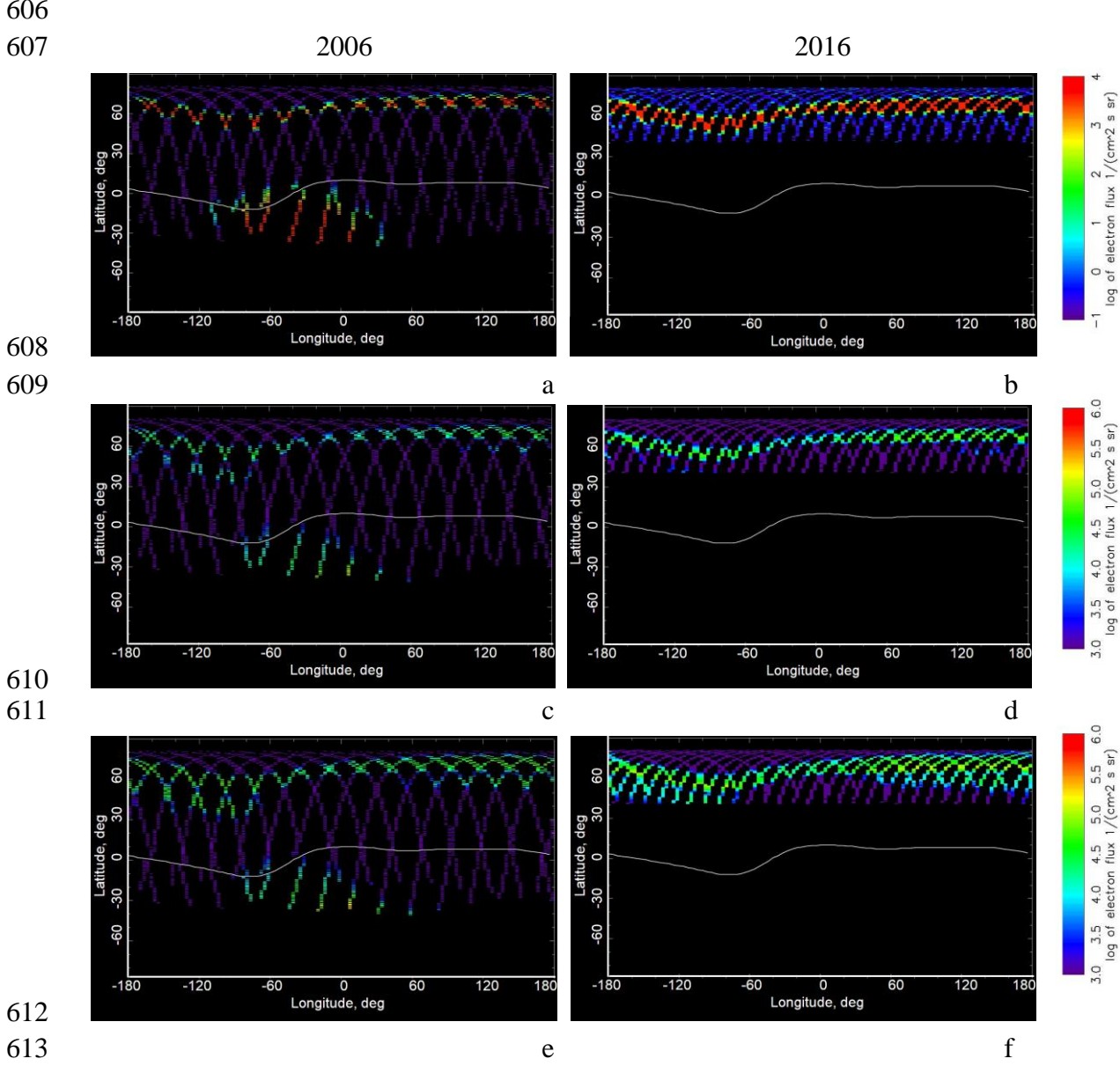

a

c

d

e

f

Figure 2. Geographic maps of energetic electron fluxes with energies >300 keV (a,b), >100 keV (c,d), >30 keV (e,f) and pitch angles of ~90° observed by POES satellites at height of ~850 km in 2 hour vicinity of local noon (left column) on 23 June 2006 and (right column) on 3 June 2016. The solid wide curve indicates the geomagnetic equator. The outer and inner electron belts and a slot region between them are clearly seen (excepting of >100 keV electrons), respectively, at high and middle latitudes in the longitudinal range from ~90° E to ~80°W.

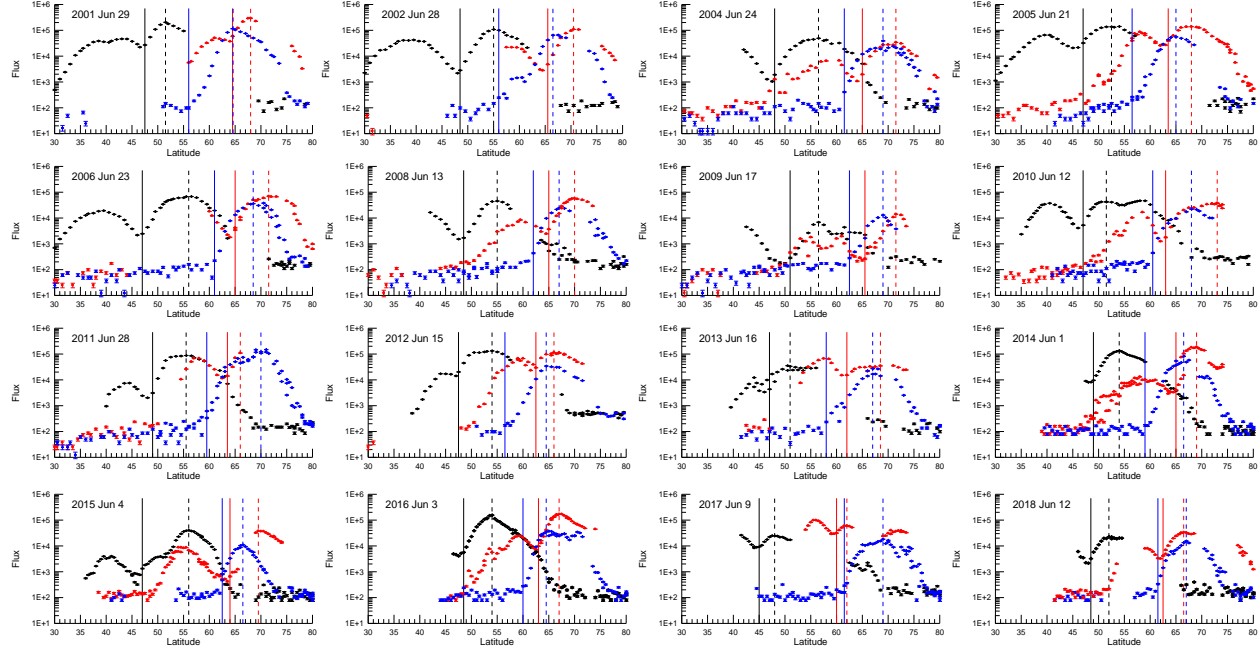

a

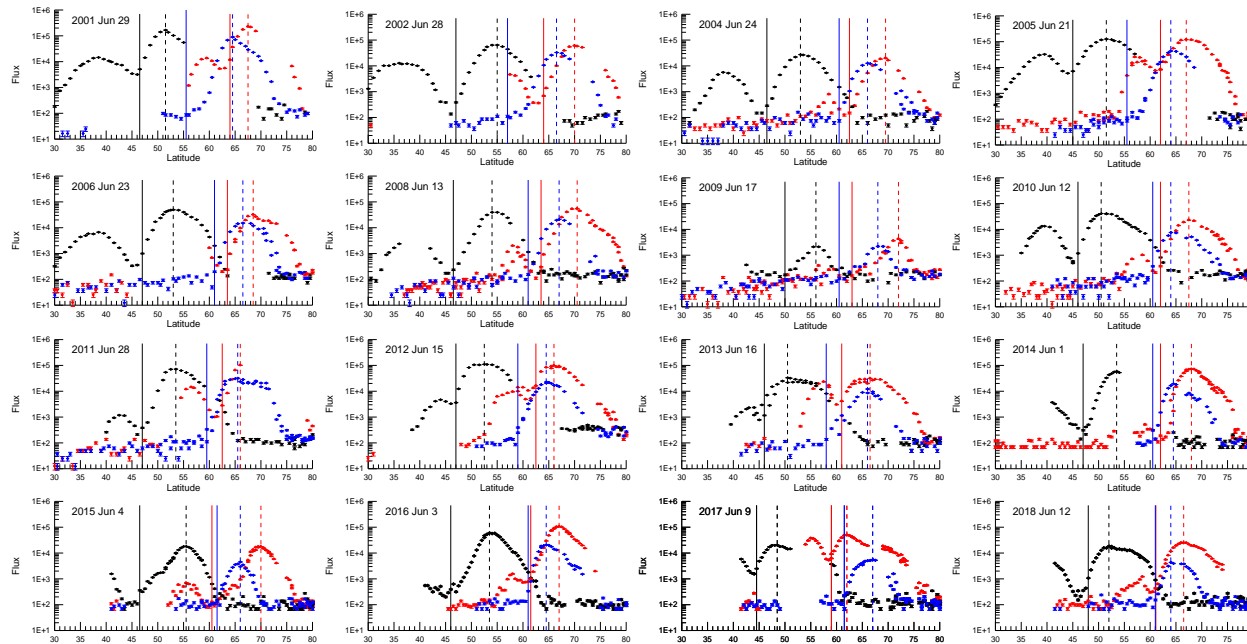

b

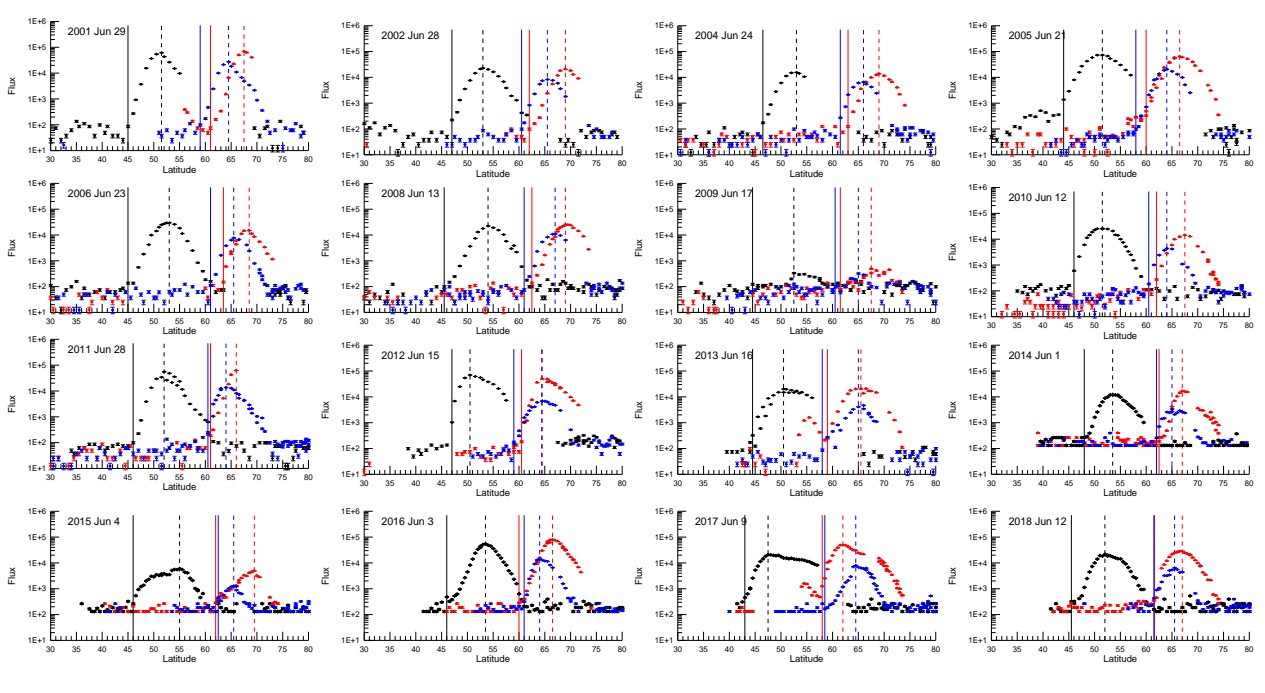

c

Figure 3. Latitudinal profiles of electron fluxes with pitch angles of ~90° observed by POES satellites during quiet days in different years at height of ~850 km in vicinity of local noon at longitudes around 100°E (red circles), 0°E (blue crosses), and 80°W (black diamonds) for various energy channels: (a) >30 keV, (b) >100 keV, and (c) >300 keV. Vertical dashed and solid lines indicate latitudes of the maximum and inner edge of the outer radiation belt, respectively.

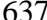

a

b

c


Figure 4. Geographic latitude of the maximum of the outer radiation belt measured at height of
~850 km during geomagnetic quiet days around 80°W (a), 0°E (b), and 100°E (c) for electrons
with energies of >30 keV (red circles), >100 keV (blue crosses), and >300 keV (green triangles).
Dashed curves of corresponding colors show the best linear fit of the latitudinal change of the
maximum location with years (see Table 3). Solid black curves show the latitudinal change
predicted by the IGRF model of corresponding epochs (see details in the text). The grey curve
shows sunspot number (right axis).


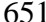

a


b


c



Figure 5. The same as Figure 4 but for the inner edge of the outer radiation belt. Coefficients of the
best linear fit are presented in Table 4.

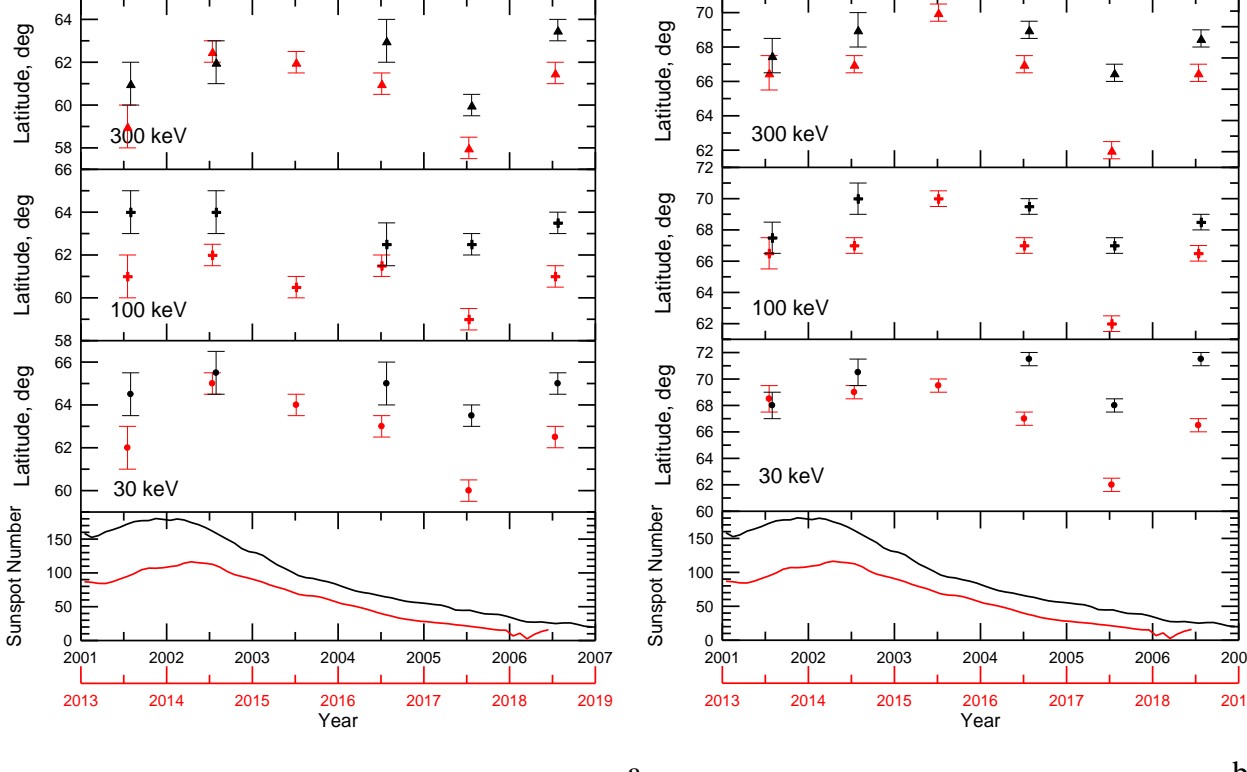

a                                                        b

Figure 6. Geographic latitude of the inner edge (a) and maximum (b) of the outer radiation belt
measured during geomagnetic quiet days at height of ~850 km around longitude of 100°E for
electrons with energies of >30 keV (circles), >100 keV (crosses), and >300 keV (triangles).
Bottom panels show the sunspot number in the 23rd solar cycle (the years 2001 – 2006, black
curves) and in the 24th solar cycle (the years 2013 – 2018, blue curves). The outer radiation belt
location is shown by black and red symbols, respectively, for the 23rd and 24th solar cycles. It can
be seen that on the declining phase of the 24th solar cycle, the outer radiation belt is systematically
located at lower latitudes than that during the 23rd solar cycle.