# Peer review of "On the radiation belt location in the 23 – 24 solar cycles"

_Annales Geophysicae, 2018_

## Referee Comment (RC1) · Anonymous Referee #1 · 20 Dec 2018

The paper addresses the variation in the outer radiation belt location during 23-24 solar cycles (years 2001-2018). The author uses the electron flux data provided by POES (Polar-Orbiting Environmental) satellites. Since outer radiation belt is highly dynamic and undergoes variations under a number of external sources, the author makes special care to exclude most of these. The residual variation of belt location corresponds, in author's opinion, to variation of geomagnetic field and solar activity. Three longitudinal sectors are investigated: 80°W (America), 0°E (Europe) and 100°E (Siberia). The main conclusion is that Siberian sector exhibits anomalous behavior: while American and European equatorial drifts of the outer radiation belt during years 2001-2018 are equal to 1° and are consistent with IGRF-12 geomagnetic field variation, the Siberian drift equals 3° and is 2° larger then predicted by IGRF-12 model. In my

opinion the paper addresses an interesting and important question how the satellite data can be used to improve geomagnetic field models. The paper has clearly stated idea, method for data processing and conclusion. However, the main conclusion of the paper seems questionable. The amount of 3° comes from linear fit of data which is too far from straight line. The author attributes the difference to variation of solar activity. In this case he has to introduce some model of solar activity influence on radiation belt position and subtract the corresponding amount from the data prior to applying linear fit. Another way is to treat variation as random, then the uncertainty in linear fit parameters should be calculated, not only their mean values. For example, linear fitting of >300 keV Siberian data for electron flux maximum with Origin 6.0 yields Lat=(-0.166±0.084)·Year+(401±169). (Here I neglected "day-of-year" shift in data, so mean values are slightly different from those in the paper.) It means that the uncertainty in the slope is about 50% and 3° may in fact reduce to 1.5°. In my opinion, the paper should be improved and resubmitted, as its main conclusion is not supported sufficiently by the data used. The author should either account explicitly for solar activity influence or calculate the uncertainty in linear fits and, probably, correct the conclusion of the paper.

---

## Author Comment (AC1) · 17 Jan 2019

I very appreciate the reviewer for very useful and valuable comments and suggestions. They help to improve substantially the quality of revised manuscript.

The following revisions have been done in according to the comments (colored by blue):
1. The using of linear fitting was justified.

2. The errors of the fitting parameters have been calculated by using a linear regression. The results are listed in new tables (Tables 3 and 4).

3. The fitting expressions have been removed from Figures 4 and 5.

4. Description of Figures 4 and 5 was substantially improved with using discussion of

the linear regression results presented in Tables 3 and 4.

5. The comparison with the IGRF model in the Siberian sector was revised in the following manner: "From Table 4, one can see that the slope a is calculated with errors of ~30% and ~20%, respectively, for >30 keV and >100 keV electrons. It means that the decrease in latitude might be ~2deg (instead of ~3deg) and ~1.5deg (instead of ~2deg), respectively. These values are again larger than 1deg of the model prediction. Hence, there is a tendency that the change in the latitudinal location of ORB maximum is underestimated by the model. This fact indicates that during 17 years from 2001 to 2018, ORB is abnormally displaced toward the lower latitudes in the Siberian sector."

6. The abstract and conclusions have been revised accordingly: "However in the Siberian sector, the model has a tendency to underestimate the equatorward shift of ORB."

Please also note the supplement to this comment:
https://www.ann-geophys-discuss.net/angeo-2018-118/angeo-2018-118-AC1-supplement.pdf

**Supplement:**

[revised manuscript text omitted]

---

## Referee Comment (RC2) · Anonymous Referee #2 · 28 Mar 2019

In their manuscript, "On the radiation belt location in the 23-24 solar cycles", Dmitriev exploited long-term data on energetic electron fluxes available from the NOAA/POES satellites that cover the time period 2001-2018 to show a clear solar cycle variation. Furthermore, the outer Van Allen belt is shown to shift equatorward as a result of the latest changes observed in the geomagnetic field that could not be predicted by the 12th generation of the IGRF model. In particular, a more than 3Î£ displacement in the geographic latitude of the maximum electron flux in the outer radiation belt over the longitudinal sector of Siberia centred around 80Î£ W – higher than in the European and North American sector - was largely attributed to sudden geomagnetic field changes that were observed during solar cycle 24 and possibly originated in the Earth's core.

The presented material offers insufficient evidence pointing towards an anomalous shift

of the outer radiation belt equatorward over the last years although electron flux data analysed have been carefully chosen so than the influence of sources of variability in the solar wind or related to geomagnetic activity are minimal. However, the solar cycle variation has not been adequately addressed. If the author can address this concern such that their conclusions are clearly supported by the data presented and can improve the placement of this work in the context of previous literature, then this manuscript could become a useful addition to the literature. Specifically, I could recommend this manuscript for publication in Annales Geophysicae subject to the specific points detailed below:

In lines 27 – 28, a brief description of the outer Van Allen radiation belt is provided where this population of charged particles is presented as part of the outer magnetosphere, contrary to what has been widely established and is presented in the following publications:

- Baker (1995), The inner magnetosphere: A review, Surveys in Geophysics, doi: 10.1007/BF01044572

- Ebihara & Miyoshi (2011), Dynamic inner magnetosphere: A tutorial and recent advances, in Liu W., Fujimoto M. (eds) The dynamic magnetosphere, doi: 10.1007/978-94-007-0501-2_9

Additional references on the long-term variations of the radiation belts' structure that should be considered are the following:

- Fung et al. (2006), Long-term variations of the electron slot region and global radiation belt structure, Geophysical Research Letters, doi: 10.1029/2005GL024891

- Baker & Kanekal (2008), Solar cycle changes, geomagnetic variations, and energetic particle properties in the inner magnetosphere, Journal of Atmospheric and Solar-Terrestrial Physics, doi: 10.1016/j.jastp.2007.08.031

- Glauert et al. (2018), A 30-year simulation of the outer electron radiation belt, Space Weather, doi: 10.1029/2018SW001981

On lines 30 and 35, the semi-annual variation of the outer radiation belt is mentioned. This seasonal and not annual change could be explained by the IMF-effect also known as Russell-McPherron effect which is described in:

- Russell & McPherron (1973), Semiannual variation of geomagnetic activity, Journal of Geophysical Research, doi: 10.1029/JA078i001p00092

- McPherron et al. (2009), Role of the Russell-McPherron effect in the acceleration of relativistic electrons, Journal of Atmospheric and Solar-Terrestrial Physics, doi: 10.1016/j.jastp.2008.11.002

and where origins of the seasonal variability in geomagnetic activity have been traced.

There are minor issues with English language use and several typographical errors.

For example, on line 54, the term auroral electrojet is first introduced that should be one word. Since Smith et al. (2017) studied both current in the north and south hemisphere, it should also be plural (auroral electrojets – AEJs).

Further down on the same page, on line 72, the work of Kataoka et al. (2015) is listed among the references for the October-November 2003 superstorms although it is focused on the magnetic storm of 17 March 2015 which is mentioned further down, in lines 74-81. It should, therefore, be moved more appropriately further down after the brief description of the 2015 Saint Patrick's Day storm.

On line 89, it is indicated that the satellite observations used for the study of the outer electron belt location cover the time period from 1998 to 2016. However, the dates listed in Table 2 are within the time period between 2001 and 2018 which is also the

time period on which this study is focused as indicated throughout the manuscript.

On line 119, it should read "geomagnetic activity was very weak" and not "very week".

On line 135, it is indicated that Figure 2 shows POES observations from 3 June 2016, while the figure caption indicated that the observations shown are from 2 June 2016.

In lines 123-126, the close link between increases in solar wind speed and enhancements in electron fluxes in the outer radiation belt is briefly described. Periodic oscillations in the Earth's magnetic field with frequencies in the range of a few millihertz (ultralow frequency waves) may indeed be an intermediary through which solar wind influences radiation belt dynamics due to their potential for resonant interactions with energetic electrons causing the radial migration of resonant electrons. It should, however, be corrected that electrons are accelerated and increase their energy when they are transported earthward to regions of stronger geomagnetic field. Recent, representative publications on this acceleration mechanism are the following:

- Mann et al. (2013), Discovery of the action of a geophysical synchrotron in the Earth's Van Allen radiation belts, Nature Communications, doi: 10.1038/ncomms3795

- Su et al. (2015), Ultra-low frequency wave-driven diffusion of radiation belt relativistic electrons, Nature Communications, doi: 10.1038/ncomms10096

Radial transport acts as a loss mechanism when particle drift outward and are lost to the magnetopause. The work of Horne et al. (2007) and Reeves et al. (2013), provided as reference, is centered on a different acceleration mechanism acting in the heart of the radiation belt (local acceleration) that involves whistler mode chorus waves rather than waves generated through the Kelvin-Helmholtz instability along the magnetopause.

In lines 187-188, the author notes that the inner edge of the outer radiation is defined as the "first high-latitude point of electron flux enhancements". Could the latitude above which the flux enhancement was searched be indicated? In addition, which criterion was applied on flux measurements to determine which fluctuations in electron flux correspond to the enhancement observed at the inner edge of the outer electron belt?

On line 203, the maximum of solar cycle 23 is indicated that it was observed in 2001 and that of solar cycle 24 in 2012-2013. It is not clear to me, and perhaps the reader, how this maximum was defined as both solar cycles were double-peaked according to the number of sunspots observed on the surface of the Sun that has been presented in Figures 4 and 5 with the solid grey line.

Further down, in the paragraph starting with line 208, an example of how the IGRF-12 model was used on the geographic coordinates of the outer radiation belt maximum flux to obtain the corresponding geomagnetic coordinates is provided. What height was selected as input to the model to determine the geomagnetic or geographic coordinates?

The choice of a simple linear fit over the set of outer radiation belt latitudes that have been calculated for the selected quiet days in the period 2001-2018 and are presented in Figures 4 and 5 puzzles me as it seems inadequate to support the main conclusion of the study. There is significant variability in the outer radiation belt location that is related to the solar activity variability that has not been accounted for – although it should - during linear regression. The difference in the variability observed in the location of the inner edge of the outer radiation belt or the location of the maximum electron flux has also not been quantified nor considered in the evaluation of the difference estimated between electron observations and magnetic field predictions from the IGRF-12.

On line 264, among the reference provided for the effect of the tilt angle variation on the location of the outer radiation belt, the study of Newell et al. (2006) is found. The specific study was centred on the cusp location as it is detailed in the next paragraph and should, therefore, be excluded from the reference list provided here.

In lines 269 and 270, the cusp location is suggested as a proxy of the outer radiation boundary. The author must imply the outer edge of the electron radiation belt or more correctly that a displacement of the cusp influences the location of the outer radiation belt but this is not clear from the text. It is also not substantiated by the findings of Newell et al. (2006). To date, the inner edge of the outer radiation belt has been suggested to be defined by the plasmapause, the outer boundary of the plasmasphere. Specifically, in the following publication:

- Baker et al. (2014), Impenetrable barrier to ultrarelativistic electrons in the Van Allen radiation belts, Nature, doi: 10.1038/nature13956

the authors analysed 20 months of electron flux data from the NASA/Van Allen Probes to identify a barrier in the inward transport of ultrarelativistic electron transport. Earlier,

- Darrouzet et al. (2013), Links between the plasmapause and the radiation belt boundaries as observed by the instruments CIS, RAPID and WHISPER onboard Cluster, Journal of Geophysical Research, doi: 10.1002/jgra.50239

had reached a different conclusion. The radiation belt location was found to be dependent on the energy range of particles examined but also that the plasmapause is more variable that the inner edge of the outer radiation belt. Namely, the inner or outer edge of the outer electron belt does not always coincide with the plasmapause.

In lines 274 and 275, the statement "Variations of the ORB location from cycle to cycle are not investigated yet" is not entirely correct. There are indeed significant limitations in such studies due to the lack of data covering several years that could be discussed at this point. Reference to studies such as Glauert et al. (2018) could also be provided.

On line 295, the findings of Finlay et al. (2015) suggesting rapid changes in the geomagnetic field in the past 15 years are briefly mentioned. Although the latest change observed in 2012-2013 seems to influence the location of the outer radiation belt, is there a signature of the change observed in 2006 and 2009 in the POES measurements from the same period analysed here?

Individual graphs in Figure 2 are difficult to read because of the dark background colour. The font size selected for the x and y axis labels are so small that, even after blowing them up to 200%, labels are still difficult to read. On the other hand, titles over the two columns (2016 for the right column and 2006 for the left column) seem to be misplaced.

Pages 24-25

Fonts on plots in Figure 3 could also be enlarged if these are selected to be the final sizes of the graphs.

---

## Author Comment (AC2) · 9 Apr 2019

I am grateful to the reviewer for very useful and valuable comments and suggestions. They help to improve substantially the quality of manuscript.

The following revisions have been done in according to the comments (colored by blue in the text):

Comment 1 In lines 27 – 28, a brief description of the outer Van Allen radiation belt is provided where this population of charged particles is presented as part of the outer magnetosphere, contrary to what has been widely established and is presented in the following publications: - Baker (1995), The inner magnetosphere: A review, Surveys in Geophysics, doi: 10.1007/BF01044572 - Ebihara & Miyoshi (2011), Dynamic inner magnetosphere: A tutorial and recent advances, in Liu W., Fujimoto M. (eds) The dynamic magnetosphere, doi: 10.1007/978-94-007-0501-2_9

Reply 1 The sentences were revised accordingly: "The outer radiation belt (ORB) is populated by energetic and relativistic electrons trapped in the magnetosphere at drift shells above L $\sim$ 3 (e.g. Ebihara and Miyoshi, 2011). The ORB is very dynamic and exhibits variations..." The term "outer magnetosphere" has been removed from whole the text.

Comment 2 Additional references on the long-term variations of the radiation belts' structure that should be considered are the following: - Fung et al. (2006), Long-term variations of the electron slot region and global radiation belt structure, Geophysical Research Letters, doi: 10.1029/2005GL024891 - Baker & Kanekal (2008), Solar cycle changes, geomagnetic variations, and energetic particle properties in the inner magnetosphere, Journal of Atmospheric and Solar-Terrestrial Physics, doi: 10.1016/j.jastp.2007.08.031 - Glauert et al. (2018), A 30-year simulation of the outer electron radiation belt, Space Weather, doi: 10.1029/2018SW001981

On lines 30 and 35, the semi-annual variation of the outer radiation belt is mentioned. This seasonal and not annual change could be explained by the IMF-effect also known as Russell-McPherron effect which is described in: - Russell & McPherron (1973), Semiannual variation of geomagnetic activity, Journal of Geophysical Research, doi: 10.1029/JA078i001p00092 - McPherron et al. (2009), Role of the Russell-McPherron effect in the acceleration of relativistic electrons, Journal of Atmospheric and Solar-Terrestrial Physics, doi: 10.1016/j.jastp.2008.11.002 and where origins of the seasonal variability in geomagnetic activity have been traced.

Reply 2 The first paragraph of Introduction section has been revised accordingly: "The ORB is very dynamic and exhibits variations in a wide temporal range: short-term storm-time and local time variations, 27-day solar rotation, seasonal and solar cycle variations (e.g. Li et al., 2001; Baker and Kanekal, 2008; Miyoshi and Kataoka, 2011).

During magnetic storms, the ORB is substantially disturbed and shifted earthward (Baker et al., 2016; Shen et al., 2017). The storm-time variation is the strongest one for both the ORB location and intensity (Baker and Kanekal, 2008). Magnetic storms produced by interplanetary coronal mass ejecta (ICME) and high-speed streams (HSS) of the solar wind from coronal holes. The seasonal variations with maxima at equinoxes can be explained by the effect of interplanetary magnetic field (IMF) orientation relative to the geomagnetic dipole (Li et al., 2001; McPherron et al., 2009). ORB manifests prominent variations with the solar cycle (Fung et al., 2006; Baker and Kanekal, 2008). It was shown that the maximum of ORB is mostly distant from the Earth in solar minimum (Miyoshi et al., 2004) and it is closest to the Earth during solar maxima (Glauert et al., 2018)."

Comment 3 There are minor issues with English language use and several typographical errors. For example, on line 54, the term auroral electrojet is first introduced that should be one word. Since Smith et al. (2017) studied both current in the north and south hemisphere, it should also be plural (auroral electrojets – AEJs).

Reply 3 Corrected

Comment 4 Further down on the same page, on line 72, the work of Kataoka et al. (2015) is listed among the references for the October-November 2003 superstorms although it is focused on the magnetic storm of 17 March 2015 which is mentioned further down, in lines 74-81. It should, therefore, be moved more appropriately further down after the brief description of the 2015 Saint Patrick's Day storm.

Reply 4 Thank you for suggestion. Corrected.

Comment 5 On line 89, it is indicated that the satellite observations used for the study of the outer electron belt location cover the time period from 1998 to 2016. However, the dates listed in Table 2 are within the time period between 2001 and 2018 which is also the time period on which this study is focused as indicated throughout the manuscript.

[Figure]

Reply 5 The year 1998 is replaced with 2001

Comment 6 On line 119, it should read "geomagnetic activity was very weak" and not "very week".

Reply 6 Corrected

Comment 7 On line 135, it is indicated that Figure 2 shows POES observations from 3 June 2016, while the figure caption indicated that the observations shown are from 2 June 2016.

Reply 7 Yes, it was misprinting. In Figure 2 caption, 2 June was replaced with 3 June.

Comment 8 In lines 123-126, the close link between increases in solar wind speed and enhancements in electron fluxes in the outer radiation belt is briefly described. Periodic oscillations in the Earth's magnetic field with frequencies in the range of a few millihertz (ultralow frequency waves) may indeed be an intermediary through which solar wind influences radiation belt dynamics due to their potential for resonant interactions with energetic electrons causing the radial migration of resonant electrons. It should, however, be corrected that electrons are accelerated and increase their energy when they are transported earthward to regions of stronger geomagnetic field. Recent, representative publications on this acceleration mechanism are the following: - Mann et al. (2013), Discovery of the action of a geophysical synchrotron in the Earth's Van Allen radiation belts, Nature Communications, doi: 10.1038/ncomms3795 - Su et al. (2015), Ultra-low frequency wave-driven diffusion of radiation belt relativistic electrons, Nature Communications, doi: 10.1038/ncomms10096 Radial transport acts as a loss mechanism when particle drift outward and are lost to the magnetopause. The work of Horne et al. (2007) and Reeves et al. (2013), provided as reference, is centered on a different acceleration mechanism acting in the heart of the radiation belt (local acceleration) that involves whistler mode chorus waves rather than waves generated through the Kelvin-Helmholtz instability along the magnetopause.

Reply 8 Thank you for very useful papers. Note that storm-time acceleration and transport of energetic electrons in ORB is not the subject of the present study, which is dealing with quiet days. Hence, this part has been revised accordingly: "Note that the solar wind with the speed of V > 400 km/s is often associated with HSSs from coronal holes. Fast solar wind streams initiate the Kelvin-Helmholtz instability at the magnetopause and also produce recurrent magnetic storms, which are accompanied by intensification of wave activity in the outer magnetosphere that results in effective acceleration and radial transport of the ORB electrons (Engebretsone et al., 1998; Tsurutani et al., 2006; Horne et al., 2007; Su et al., 2015)."

Comment 9 In lines 187-188, the author notes that the inner edge of the outer radiation is defined as the "first high-latitude point of electron flux enhancements". Could the latitude above which the flux enhancement was searched be indicated? In addition, which criterion was applied on flux measurements to determine which fluctuations in electron flux correspond to the enhancement observed at the inner edge of the outer electron belt?

Reply 9 This important issue is described in more details in the revised manuscript: "Apparently, the electron flux enhancements peak in the maximum of ORB. Hence, the inner edge of ORB corresponds to the beginning of continuous increase of the electron flux from the minimum at low latitudes to the ORB maximum. This criterion allows determining of the inner edge for the electrons with energies >300 keV and in the European sector, where the slot region is not so obvious. Geographic latitude of the inner edge is determined for each year with the accuracy varying from 0.5ïĆř to 1ïĆř. In the American sector, the inner edge of ORB is situated at lowest latitudes from 43ïĆř to 51ïĆř, in the European sector – from 55ïĆř to 63ïĆř, and in the Siberian sector – at highest latitudes from 58ïĆř to 65ïĆř."

Comment 10 On line 203, the maximum of solar cycle 23 is indicated that it was observed in 2001 and that of solar cycle 24 in 2012-2013. It is not clear to me, and perhaps the reader, how this maximum was defined as both solar cycles were doublepeaked according to the number of sunspots observed on the surface of the Sun that has been presented in Figures 4 and 5 with the solid grey line.

Reply 10 Solar maximum, as a physical phenomenon of the solar magnetic field reversal, has the double-peak structure both in the 23rd and 24th solar cycles. Those cycles peaked, respectively, in November 2001 and in April 2014. After the peaks, the declining phases started and the solar activity was quickly decreasing. Dramatic changes in the solar magnetic field (including the reversals) were observed from 2000 to 2001 and from 2012 to the beginning of 2014, respectively. In this sense, the maximal phases of those cycles occurred definitely in 2001 - 2002 and 2012 - 2013. The year of 2014 belongs both to the maximum and to the declining phase of the 24th solar cycle. In the original paper, the year 2000 was not shown and, hence, it was mentioned. In the revised manuscript, this issue is described in more details: "Note that the maximum phases of the 23rd and 24th solar cycles occurred in the years 2000 - 2001 and in 2012 – April 2014, respectively. The years 2008 – 2009 are the solar minimum phase. The declining phases lasted from 2003 to 2007 and from 2014 to 2018. In Figures 4 and 5, one can see that during the declining phase of the current 24th solar cycle (especially in the years 2016 – 2018), the behavior of the ORB maximum and inner edge is different from that during the declining phase of the previous 23rd solar cycle. Namely, their latitudes increased only slightly or even decreased above North America and especially above Siberia."

Comment 11 Further down, in the paragraph starting with line 208, an example of how the IGRF-12 model was used on the geographic coordinates of the outer radiation belt maximum flux to obtain the corresponding geomagnetic coordinates is provided. What height was selected as input to the model to determine the geomagnetic or geographic coordinates?

Reply 11 The height of POES orbit at 850 km was used. Actually for a given magnetic coordinates, the long-term changes in geographic latitudes predicted by the IGRF-12 model do not vary much with the heights for the low-earth orbits and ground (10% of
the Earth's radius). The latitudes are different, but their changes with time are almost same.

Comment 12 The choice of a simple linear fit over the set of outer radiation belt latitudes that have been calculated for the selected quiet days in the period 2001-2018 and are presented in Figures 4 and 5 puzzles me as it seems inadequate to support the main conclusion of the study. There is significant variability in the outer radiation belt location that is related to the solar activity variability that has not been accounted for – although it should - during linear regression.

Reply 12 The using of linear fit can be justified in the following way: "Unfortunately, there is no any model of the ORB location variation with the solar cycle because the driving mechanisms are not well established. As a first approach, the variations of ORB location with years are considered as random and can be fitted by a linear function (indicated by dashed strait lines in Figures 4 and 5): ïĄň = a * year + b, (1) where ïĄň is the latitude of maximum or inner edge of ORB. The slope a, parameter b and their standard errors are calculated from a linear regression for various longitudinal regions and various energies of electrons. The results are presented in Tables 3 and 4 for the ORB maximum and the inner edge, respectively."

Further "As one can see in Figures 4 and 5, the long-term variation in IGRF-12 is almost linear function of the year and, hence, this variation can be easily compared with the linear fits of the ORB location."

Comment 13 The difference in the variability observed in the location of the inner edge of the outer radiation belt or the location of the maximum electron flux has also not been quantified nor considered in the evaluation of the difference estimated between electron observations and magnetic field predictions from the IGRF-12.

Reply 13 An effort of inter comparison between the ORB inner edge and maximum was made in the original paper: "As can be seen in Figures 4 and 5, the location of ORB manifests the well-known solar cycle variation: the latitudes of ORB maximum and inner edge have a tendency to be highest around solar minimum in 2008 – 2009 and lowest during solar maxima in the years 2001 and 2012 – 2013." In the revised manuscript, the difference is quantified and discussed for each longitudinal sector. For example: "Similar pattern can be found for the inner edge of ORB in the Siberian sector (see Figure 5c). Namely, the IGRF model predicts a decrease of ∼1ïĆř. The inner edge was shifted toward lower latitudes by ∼3ïĆř, ∼2ïĆř and ∼1ïĆř, respectively, for >30 keV, >100 keV and >300 keV electrons. From Table 4, one can see that the slope a is calculated with errors of ∼30% and ∼20%, respectively, for >30 keV and >100 keV electrons. It means that the decrease in latitude might be ∼2ïĆř (instead of ∼3ïĆř) and ∼1.5ïĆř (instead of ∼2ïĆř), respectively. These values are again larger than 1ïĆř of the model prediction. Hence, there is a tendency that the change in the latitudinal location of ORB maximum is underestimated by the model. This fact indicates that during 17 years from 2001 to 2018, ORB is abnormally displaced toward the lower latitudes in the Siberian sector."

Comment 14 On line 264, among the reference provided for the effect of the tilt angle variation on the location of the outer radiation belt, the study of Newell et al. (2006) is found. The specific study was centred on the cusp location as it is detailed in the next paragraph and should, therefore, be excluded from the reference list provided here.

Reply 14 Corrected, the reference has been replaced.

Comment 15 In lines 269 and 270, the cusp location is suggested as a proxy of the outer radiation boundary. The author must imply the outer edge of the electron radiation belt or more correctly that a displacement of the cusp influences the location of the outer radiation belt but this is not clear from the text. It is also not substantiated by the findings of Newell et al. (2006). To date, the inner edge of the outer radiation belt has been suggested to be defined by the plasmapause, the outer boundary of the plasmasphere. Specifically, in the following publication: - Baker et al. (2014), Impenetrable barrier to ultrarelativistic electrons in the Van Allen radiation belts, Nature, doi: 10.1038/nature13956 the authors analysed 20 months of electron flux data from the

[Figure]

NASA/Van Allen Probes to identify a barrier in the inward transport of ultrarelativistic electron transport. Earlier, - Darrouzet et al. (2013), Links between the plasmapause and the radiation belt boundaries as observed by the instruments CIS, RAPID and WHISPER onboard Cluster, Journal of Geophysical Research, doi: 10.1002/jgra.50239 had reached a different conclusion. The radiation belt location was found to be dependent on the energy range of particles examined but also that the plasmapause is more variable that the inner edge of the outer radiation belt. Namely, the inner or outer edge of the outer electron belt does not always coincide with the plasmapause.

Reply 15 Thank you very much for very useful papers. This paragraph has been revised accordingly: "The effect of solar wind parameters, including IMF Bz and dynamic pressure (Pd), to the ORB location is not obvious. It is found that the slot region location can be related to the plasmapause but the relation is ambiguous (Darrouzet et al., 2013; Baker et al., 2014). We can make indirect estimation of the effect using a dependence of the cusp location from the solar wind parameters (Kuznetsov et al., 1993; Newell et al., 2006). The equatorward edge of the cusp separates the open and close magnetic filed lines in the dayside magnetosphere. Hence the latitude of the equatorward edge can be considered as a proxy of the ORB outer edge. In the first approach, we assume that the effect of solar wind parameters to the ORB location can be represented by the dynamics of the ORB outer edge or the cusp equatorward edge. It can be shown that Bz = -4 nT results in less than 0.5ïĆř equatorward shift of the cusp and a change of Pd from 1 to 2 nPa results in ∼0.2ïĆř decrease in the latitude of the cusp equatorward edge. Hence, the effects of both Pd and IMF Bz are several times weaker than the difference of 3ïĆř."

Comment 16 In lines 274 and 275, the statement "Variations of the ORB location from cycle to cycle are not investigated yet" is not entirely correct. There are indeed significant limitations in such studies due to the lack of data covering several years that could be discussed at this point. Reference to studies such as Glauert et al. (2018) could also be provided.

Reply 16 Thank you very much for very useful paper. This part was revised accordingly: "Variations of the ORB location from cycle to cycle and during different phases of solar cycles are still poorly investigated. It was well established that during solar minima and maxima, the ORB is located, respectively, at highest and lowest latitudes (Miyoshi et al., 2004; Glauert et al., 2018). From these findings, we can speculate that lower(higher) solar activity results in an increase (a decrease) of the ORB latitudes."

Comment 17 On line 295, the findings of Finlay et al. (2015) suggesting rapid changes in the geomagnetic field in the past 15 years are briefly mentioned. Although the latest change observed in 2012-2013 seems to influence the location of the outer radiation belt, is there a signature of the change observed in 2006 and 2009 in the POES measurements from the same period analysed here?

Reply 17 This important issue is clarified in the following manner: "We can assume that the abnormal ORB displacement might be related to the geomagnetic jerks. Though, there is no prominent change in the ORB location in 2006, one can indicate very high latitude of ORB in 2009. Note that the jerk in 2009 coincided with the abnormally deep solar minimum and, hence, it could be hard to distinguish between the two effects. On the other hand, we have found significant change in the ORB dynamics after 2012 – 2013."

Comment 18 Individual graphs in Figure 2 are difficult to read because of the dark background colour. The font size selected for the x and y axis labels are so small that, even after blowing them up to 200%, labels are still difficult to read. On the other hand, titles over the two columns (2016 for the right column and 2006 for the left column) seem to be misplaced.

Reply 18 Figure 2 has been revised accordingly.

Comment 19 Fonts on plots in Figure 3 could also be enlarged if these are selected to be the final sizes of the graphs.

Reply 19 The multi-plot Figure 3 is mainly presented in order to demonstrate qualitatively the structure of ORB and its dynamics in various longitudinal sectors. The results of numerical analysis are presented in Figures 4 and 5.

Please also note the supplement to this comment:
https://www.ann-geophys-discuss.net/angeo-2018-118/angeo-2018-118-AC2-supplement.pdf

**Supplement:**

**Reply to Reviewer's comments.**

I am grateful to the reviewer for very useful and valuable comments and suggestions. They help to improve substantially the quality of manuscript.

The following revisions have been done in according to the comments (colored by blue in the text):

Comment 1

*In lines 27 – 28, a brief description of the outer Van Allen radiation belt is provided where this population of charged particles is presented as part of the outer magnetosphere, contrary to what has been widely established and is presented in the following publications:*

*- Baker (1995), The inner magnetosphere: A review, Surveys in Geophysics, doi: 10.1007/BF01044572*

*- Ebihara & Miyoshi (2011), Dynamic inner magnetosphere: A tutorial and recent advances,*

*in Liu W., Fujimoto M. (eds) The dynamic magnetosphere, doi: 10.1007/978-94-007-0501-2_9*

Reply 1

The sentences were revised accordingly:

"The outer radiation belt (ORB) is populated by energetic and relativistic electrons trapped in the magnetosphere at drift shells above $L \sim 3$ (e.g. Ebihara and Miyoshi, 2011). The ORB is very dynamic and exhibits variations…"

The term "outer magnetosphere" has been removed from whole the text.

Comment 2

*Additional references on the long-term variations of the radiation belts' structure that should be considered are the following:*

*- Fung et al. (2006), Long-term variations of the electron slot region and global radiation belt structure, Geophysical Research Letters, doi: 10.1029/2005GL024891*

*- Baker & Kanekal (2008), Solar cycle changes, geomagnetic variations, and energetic particle properties in the inner magnetosphere, Journal of Atmospheric and Solar-Terrestrial Physics, doi: 10.1016/j.jastp.2007.08.031*

*- Glauert et al. (2018), A 30-year simulation of the outer electron radiation belt, Space Weather, doi: 10.1029/2018SW001981*

*On lines 30 and 35, the semi-annual variation of the outer radiation belt is mentioned. This seasonal and not annual change could be explained by the IMF-effect also known as Russell-McPherron effect which is described in:*

*- Russell & McPherron (1973), Semiannual variation of geomagnetic activity, Journal of Geophysical Research, doi: 10.1029/JA078i001p00092*

*- McPherron et al. (2009), Role of the Russell-McPherron effect in the acceleration of relativistic electrons, Journal of Atmospheric and Solar-Terrestrial Physics, doi: 10.1016/j.jastp.2008.11.002*

*and where origins of the seasonal variability in geomagnetic activity have been traced.*

Reply 2

The first paragraph of Introduction section has been revised accordingly:

"The ORB is very dynamic and exhibits variations in a wide temporal range: short-term storm-time and local time variations, 27-day solar rotation, seasonal and solar cycle variations (e.g. Li et al.,

2001; Baker and Kanekal, 2008; Miyoshi and Kataoka, 2011). During magnetic storms, the ORB is substantially disturbed and shifted earthward (Baker et al., 2016; Shen et al., 2017). The storm-time variation is the strongest one for both the ORB location and intensity (Baker and Kanekal, 2008).

Magnetic storms produced by interplanetary coronal mass ejecta (ICME) and high-speed streams (HSS) of the solar wind from coronal holes. The seasonal variations with maxima at equinoxes can be explained by the effect of interplanetary magnetic field (IMF) orientation relative to the geomagnetic dipole (Li et al., 2001; McPherron et al., 2009). ORB manifests prominent variations with the solar cycle (Fung et al., 2006; Baker and Kanekal, 2008). It was shown that the maximum of ORB is mostly distant from the Earth in solar minimum (Miyoshi et al., 2004) and it is closest to the Earth during solar maxima (Glauert et al., 2018)."

Comment 3

*There are minor issues with English language use and several typographical errors.*

*For example, on line 54, the term auroral electrojet is first introduced that should be one word. Since Smith*

*et al. (2017) studied both current in the north and south hemisphere, it should also be plural (auroral*

*electrojets – AEJs).*

Reply 3

Corrected

Comment 4

*Further down on the same page, on line 72, the work of Kataoka et al. (2015) is listed among the references*

*for the October-November 2003 superstorms although it is focused on the magnetic storm of 17 March*

*2015 which is mentioned further down, in lines 74-81. It should, therefore, be moved more appropriately*

*further down after the brief description of the 2015 Saint Patrick's Day storm.*

Reply 4

Thank you for suggestion. Corrected.

Comment 5

*On line 89, it is indicated that the satellite observations used for the study of the outer electron belt location*

*cover the time period from 1998 to 2016. However, the dates listed in Table 2 are within the time period*

*between 2001 and 2018 which is also the time period on which this study is focused as indicated throughout*

*the manuscript.*

Reply 5

The year 1998 is replaced with 2001

Comment 6

*On line 119, it should read "geomagnetic activity was very weak" and not "very week".*

Reply 6

Corrected

Comment 7

*On line 135, it is indicated that Figure 2 shows POES observations from 3 June 2016, while the figure*

*caption indicated that the observations shown are from 2 June 2016.*

Reply 7

Yes, it was misprinting. In Figure 2 caption, 2 June was replaced with 3 June.

Comment 8

*In lines 123-126, the close link between increases in solar wind speed and enhancements in electron fluxes*

*in the outer radiation belt is briefly described. Periodic oscillations in the Earth's magnetic field with*

*frequencies in the range of a few millihertz (ultralow frequency waves) may indeed be an intermediary*

*through which solar wind influences radiation belt dynamics due to their potential for resonant interactions*

*with energetic electrons causing the radial migration of resonant electrons. It should, however, be corrected*

*that electrons are accelerated and increase their energy when they are transported earthward to regions of*

*stronger geomagnetic field. Recent, representative publications on this acceleration mechanism are the*

*following:*

*- Mann et al. (2013), Discovery of the action of a geophysical synchrotron in the Earth's Van Allen radiation*

*belts, Nature Communications, doi: 10.1038/ncomms3795*

*- Su et al. (2015), Ultra-low frequency wave-driven diffusion of radiation belt relativistic electrons, Nature*

*Communications, doi: 10.1038/ncomms10096*

*Radial transport acts as a loss mechanism when particle drift outward and are lost to the magnetopause.*

*The work of Horne et al. (2007) and Reeves et al. (2013), provided as reference, is centered on a different*

*acceleration mechanism acting in the heart of the radiation belt (local acceleration) that involves whistler*

*mode chorus waves rather than waves generated through the Kelvin-Helmholtz instability along the*

*magnetopause.*

Reply 8

Thank you for very useful papers. Note that storm-time acceleration and transport of energetic electrons in
ORB is not the subject of the present study, which is dealing with quiet days. Hence, this part has been
revised accordingly:
"Note that the solar wind with the speed of V > 400 km/s is often associated with HSSs from
coronal holes. Fast solar wind streams initiate the Kelvin-Helmholtz instability at the
magnetopause and also produce recurrent magnetic storms, which are accompanied by
intensification of wave activity in the outer magnetosphere that results in effective acceleration and
radial transport of the ORB electrons (Engebretsone et al., 1998; Tsurutani et al., 2006; Horne et al.,
2007; Su et al., 2015)."
Comment 9
*In lines 187-188, the author notes that the inner edge of the outer radiation is defined as the "first*
*high-latitude point of electron flux enhancements". Could the latitude above which the flux enhancement*
*was searched be indicated? In addition, which criterion was applied on flux measurements to determine*
*which fluctuations in electron flux correspond to the enhancement observed at the inner edge of the outer*
*electron belt?*
Reply 9
This important issue is described in more details in the revised manuscript:
"Apparently, the electron flux enhancements peak in the maximum of ORB. Hence, the inner edge
of ORB corresponds to the beginning of continuous increase of the electron flux from the
minimum at low latitudes to the ORB maximum. This criterion allows determining of the inner
edge for the electrons with energies >300 keV and in the European sector, where the slot region is
not so obvious. Geographic latitude of the inner edge is determined for each year with the accuracy
varying from 0.5° to 1°. In the American sector, the inner edge of ORB is situated at lowest
latitudes from 43° to 51°, in the European sector – from 55° to 63°, and in the Siberian sector – at
highest latitudes from 58° to 65°."
Comment 10
*On line 203, the maximum of solar cycle 23 is indicated that it was observed in 2001 and that of solar cycle*
*24 in 2012-2013. It is not clear to me, and perhaps the reader, how this maximum was defined as both solar*
*cycles were double-peaked according to the number of sunspots observed on the surface of the Sun that has*
*been presented in Figures 4 and 5 with the solid grey line.*
Reply 10
Solar maximum, as a physical phenomenon of the solar magnetic field reversal, has the double-peak
structure both in the 23rd and 24th solar cycles. Those cycles peaked, respectively, in November 2001 and in
April 2014. After the peaks, the declining phases started and the solar activity was quickly decreasing.
Dramatic changes in the solar magnetic field (including the reversals) were observed from 2000 to 2001 and from 2012 to the beginning of 2014, respectively. In this sense, the maximal phases of those cycles occurred definitely in 2001 - 2002 and 2012 - 2013. The year of 2014 belongs both to the maximum and to the declining phase of the 24$^{th}$ solar cycle.

In the original paper, the year 2000 was not shown and, hence, it was mentioned.

In the revised manuscript, this issue is described in more details:

"Note that the maximum phases of the 23$^{rd}$ and 24$^{th}$ solar cycles occurred in the years 2000 - 2001

and in 2012 – April 2014, respectively. The years 2008 – 2009 are the solar minimum phase. The declining phases lasted from 2003 to 2007 and from 2014 to 2018. In Figures 4 and 5, one can see that during the declining phase of the current 24$^{th}$ solar cycle (especially in the years 2016 – 2018), the behavior of the ORB maximum and inner edge is different from that during the declining phase of the previous 23$^{rd}$ solar cycle. Namely, their latitudes increased only slightly or even decreased above North America and especially above Siberia."

Comment 11

*Further down, in the paragraph starting with line 208, an example of how the IGRF-12 model was used on*

*the geographic coordinates of the outer radiation belt maximum flux to obtain the corresponding*

*geomagnetic coordinates is provided. What height was selected as input to the model to determine the*

*geomagnetic or geographic coordinates?*

Reply 11

The height of POES orbit at 850 km was used. Actually for a given magnetic coordinates, the long-term changes in geographic latitudes predicted by the IGRF-12 model do not vary much with the heights for the low-earth orbits and ground (10% of the Earth's radius). The latitudes are different, but their changes with time are almost same.

Comment 12

*The choice of a simple linear fit over the set of outer radiation belt latitudes that have been calculated for*

*the selected quiet days in the period 2001-2018 and are presented in Figures 4 and 5 puzzles me as it seems*

*inadequate to support the main conclusion of the study. There is significant variability in the outer radiation*

*belt location that is related to the solar activity variability that has not been accounted for – although it*

*should - during linear regression.*

Reply 12

The using of linear fit can be justified in the following way:

"Unfortunately, there is no any model of the ORB location variation with the solar cycle because the driving mechanisms are not well established.

As a first approach, the variations of ORB location with years are considered as random and can be fitted by a linear function (indicated by dashed strait lines in Figures 4 and 5):

$$\lambda = a * \text{year} + b, \text{ (1)}$$

where $\lambda$ is the latitude of maximum or inner edge of ORB. The slope $a$, parameter $b$ and their standard errors are calculated from a linear regression for various longitudinal regions and various energies of electrons. The results are presented in Tables 3 and 4 for the ORB maximum and the inner edge, respectively."

Further

"As one can see in Figures 4 and 5, the long-term variation in IGRF-12 is almost linear function of the year and, hence, this variation can be easily compared with the linear fits of the ORB location."

Comment 13

*The difference in the variability observed in the location of the inner edge of the outer radiation belt or the*

*location of the maximum electron flux has also not been quantified nor considered in the evaluation of the*

*difference estimated between electron observations and magnetic field predictions from the IGRF-12.*

Reply 13

An effort of inter comparison between the ORB inner edge and maximum was made in the original paper:

"As can be seen in Figures 4 and 5, the location of ORB manifests the well-known solar cycle variation: the latitudes of ORB maximum and inner edge have a tendency to be highest around solar minimum in 2008 – 2009 and lowest during solar maxima in the years 2001 and 2012 –

2013."

In the revised manuscript, the difference is quantified and discussed for each longitudinal sector. For example:

"Similar pattern can be found for the inner edge of ORB in the Siberian sector (see Figure 5c).

Namely, the IGRF model predicts a decrease of ~1°. The inner edge was shifted toward lower latitudes by ~3°, ~2° and ~1°, respectively, for >30 keV, >100 keV and >300 keV electrons. From

Table 4, one can see that the slope $a$ is calculated with errors of ~30% and ~20%, respectively, for

>30 keV and >100 keV electrons. It means that the decrease in latitude might be ~2° (instead of

~3°) and ~1.5° (instead of ~2°), respectively. These values are again larger than 1° of the model prediction. Hence, there is a tendency that the change in the latitudinal location of ORB maximum is underestimated by the model. This fact indicates that during 17 years from 2001 to 2018, ORB is abnormally displaced toward the lower latitudes in the Siberian sector."

Comment 14

*On line 264, among the reference provided for the effect of the tilt angle variation on the location of the*

*outer radiation belt, the study of Newell et al. (2006) is found. The specific study was centred on the cusp*

*location as it is detailed in the next paragraph and should, therefore, be excluded from the reference list*

*provided here.*

Reply 14

Corrected, the reference has been replaced.

Comment 15

*In lines 269 and 270, the cusp location is suggested as a proxy of the outer radiation boundary. The author*

*must imply the outer edge of the electron radiation belt or more correctly that a displacement of the cusp*

*influences the location of the outer radiation belt but this is not clear from the text. It is also not*

*substantiated by the findings of Newell et al. (2006).*

*To date, the inner edge of the outer radiation belt has been suggested to be defined by the plasmapause, the*

*outer boundary of the plasmasphere. Specifically, in the following publication:*

*- Baker et al. (2014), Impenetrable barrier to ultrarelativistic electrons in the Van Allen radiation belts,*

*Nature, doi: 10.1038/nature13956 the authors analysed 20 months of electron flux data from the NASA/Van*

*Allen Probes to identify a barrier in the inward transport of ultrarelativistic electron transport.*

*Earlier, - Darrouzet et al. (2013), Links between the plasmapause and the radiation belt boundaries as*

*observed by the instruments CIS, RAPID and WHISPER onboard Cluster, Journal of Geophysical Research,*

*doi: 10.1002/jgra.50239*

*had reached a different conclusion. The radiation belt location was found to be dependent on the energy*

*range of particles examined but also that the plasmapause is more variable that the inner edge of the outer*

*radiation belt. Namely, the inner or outer edge of the outer electron belt does not always coincide with the*

*plasmapause.*

Reply 15

Thank you very much for very useful papers.

This paragraph has been revised accordingly:

"The effect of solar wind parameters, including IMF $Bz$ and dynamic pressure ($Pd$), to the ORB

location is not obvious. It is found that the slot region location can be related to the plasmapause but the relation is ambiguous (Darrouzet et al., 2013; Baker et al., 2014). We can make indirect estimation of the effect using a dependence of the cusp location from the solar wind parameters (Kuznetsov et al., 1993; Newell et al., 2006). The equatorward edge of the cusp separates the open and close magnetic filed lines in the dayside magnetosphere. Hence the latitude of the equatorward edge can be considered as a proxy of the ORB outer edge. In the first approach, we assume that the effect of solar wind parameters to the ORB location can be represented by the dynamics of the

ORB outer edge or the cusp equatorward edge. It can be shown that $Bz$ = -4 nT results in less than

0.5° equatorward shift of the cusp and a change of $Pd$ from 1 to 2 nPa results in ~0.2° decrease in the latitude of the cusp equatorward edge. Hence, the effects of both $Pd$ and IMF Bz are several times weaker than the difference of 3°."

Comment 16

*In lines 274 and 275, the statement "Variations of the ORB location from cycle to cycle are not investigated*

*yet" is not entirely correct. There are indeed significant limitations in such studies due to the lack of data*

*covering several years that could be discussed at this point. Reference to studies such as Glauert et al.*

*(2018) could also be provided.*

Reply 16

Thank you very much for very useful paper. This part was revised accordingly:

"Variations of the ORB location from cycle to cycle and during different phases of solar cycles are still poorly investigated. It was well established that during solar minima and maxima, the ORB is located, respectively, at highest and lowest latitudes (Miyoshi et al., 2004; Glauert et al., 2018).

From these findings, we can speculate that lower(higher) solar activity results in an increase (a decrease) of the ORB latitudes."

Comment 17

*On line 295, the findings of Finlay et al. (2015) suggesting rapid changes in the geomagnetic field in the*

*past 15 years are briefly mentioned. Although the latest change observed in 2012-2013 seems to influence*

*the location of the outer radiation belt, is there a signature of the change observed in 2006 and 2009 in the*

*POES measurements from the same period analysed here?*

Reply 17

This important issue is clarified in the following manner:

"We can assume that the abnormal ORB displacement might be related to the geomagnetic jerks.

Though, there is no prominent change in the ORB location in 2006, one can indicate very high latitude of

ORB in 2009. Note that the jerk in 2009 coincided with the abnormally deep solar minimum and, hence, it could be hard to distinguish between the two effects. On the other hand, we have found significant change in the ORB dynamics after 2012 – 2013."

Comment 18

*Individual graphs in Figure 2 are difficult to read because of the dark background colour.*

*The font size selected for the x and y axis labels are so small that, even after blowing them up to 200%,*

*labels are still difficult to read. On the other hand, titles over the two columns (2016 for the right column*

*and 2006 for the left column) seem to be misplaced.*

Reply 18

Figure 2 has been revised accordingly.

Comment 19

*Fonts on plots in Figure 3 could also be enlarged if these are selected to be the final sizes of the graphs.*

Reply 19

The multi-plot Figure 3 is mainly presented in order to demonstrate qualitatively the structure of

ORB and its dynamics in various longitudinal sectors. The results of numerical analysis are presented in Figures 4 and 5.

[revised manuscript text omitted]
. Hence, the inner edge of ORB corresponds to the beginning of continuous increase of the electron flux from the minimum at low latitudes to the ORB maximum. This criterion allows determining of the inner edge for the electrons with energies >300 keV and in the European sector, where the slot region is not so obvious. Geographic latitude of the inner edge is determined for each year with the accuracy varying from 0.5° to 1°. In the American sector, the inner edge of ORB is situated at lowest latitudes from 43° to 51°, in the European sector – from 55° to 63°, and in the Siberian sector – at highest latitudes from 58° to 65°. In Figure 3, one can find that the latitude of ORB edge above Siberia decreases with years from ~65° to 60° for all energy range of electrons. The change of ORB location above the Europe and North America is not so obvious.

Figures 4 and Figure 5 show long-term variations in the location of ORB and corresponding predictions of the IGRF-12 model during 17 years from 2001 to 2018. As one can see, the ORB maximum and inner edge of >30 keV electrons are usually located at higher latitudes than those of >100 keV electrons, and the ORB of subrelativistic >300 keV electrons is located at lowest latitudes. Note that the location of ORB maximum for >30 keV electrons is scattered significantly and it is different from those for the more energetic electrons because of substantial contamination from the auroral electrons. In contrast, the ORB maxima and inner edge of >100 keV and >300 keV electrons demonstrate very similar dynamics.

As can be seen in Figures 4 and 5, the location of ORB manifests the well-known solar cycle variation: the latitudes of ORB maximum and inner edge have a tendency to be highest around solar minimum in 2008 – 2009 and lowest during solar maxima in the years 2001 and 2012 – 2013. Note that the maximum phases of the 23rd and 24th solar cycles occurred in the years 2000 - 2001 and in 2012 – April 2014, respectively. The years 2008 – 2009 are the solar minimum phase. The declining phases lasted from 2003 to 2007 and from 2014 to 2018. In Figures 4 and 5, one can see that during the declining phase of the current 24th solar cycle (especially in the years 2016 – 2018), the behavior of the ORB maximum and inner edge is different from that during the declining phase of the previous 23rd solar cycle. Namely, their latitudes increased only slightly or even decreased above North America and especially above Siberia. Unfortunately, there is no any model of the ORB location variation with the solar cycle because the driving mechanisms are not well established.

As a first approach, the variations of ORB location with years are considered as random and can be fitted by a linear function (indicated by dashed strait lines in Figures 4 and 5):

$$\lambda = a * \text{year} + b, \text{ (1)}$$

where $\lambda$ is the latitude of maximum or inner edge of ORB. The slope $a$, parameter $b$ and their standard errors are calculated from a linear regression for various longitudinal regions and various energies of electrons. The results are presented in Tables 3 and 4 for the ORB maximum and the inner edge, respectively.

The linear fits are compared with geomagnetic field trends predicted by the IGRF model in different regions. The trends were calculated in the following manner. First, we took a point with given geographic coordinates and calculated its magnetic coordinates for the quiet day on 29 June 2001 using the IGRF model of epoch 2000. Namely, for the ORB maximum, we took points (70°N, 80°W), (66°N, 0°E) and (54°N, 100°E), respectively, for the American, European and Siberian sectors and calculated their geomagnetic coordinates (64.12°N, 11.44°W), (67.05°N, 95.66°E) and (59.5°N, 174.3°E), respectively. For the inner edge of ORB, we took, respectively, (46.5°N, 80°W), (59°N, 0°E) and (63°N, 100°E), with corresponding geomagnetic coordinates (56.62°N, 10.61°W) (60.59°N, 89.34°E) and (52.47°N, 173.7°E). Then we supposed that the geomagnetic coordinates of the points do not change with time and we used them to calculate geographic coordinates from the IGRF-12 model for corresponding quiet days listed in Table 2. The geographic coordinates of a point with given magnetic coordinates should be changed with time because of long-term variation of the geomagnetic field. As one can see in Figures 4 and 5, the long-term variation in IGRF-12 is almost linear function of the year and, hence, this variation can be easily compared with the linear fits of the ORB location.

[revised manuscript text omitted]

---

## Author Response (AR1)

**Reply to Reviewer's #1 comments.**

I very appreciate the reviewer for very useful and valuable comments and suggestions. They help to improve substantially the quality of revised manuscript.

The following revisions have been done in according to the comments (colored by blue):

1. The using of linear fitting was justified.

"Unfortunately, there is no any model of the ORB location variation with the solar cycle because the driving mechanisms are not well established. On the other hand, the long-term variation in IGRF-12 is almost linear function of the year, as one can see in Figures 4 and 5. Hence, as a first approach for comparative analysis, the variations of ORB location with years are considered as random around a linear function (indicated by dashed strait lines in Figures 4 and 5):

$$\lambda = a * year + b, (1)$$

where $\lambda$ is the latitude of maximum or inner edge of ORB. The slope $a$, parameter $b$ and their standard errors are calculated from a linear regression for various longitudinal regions and various energies of electrons. The results are presented in Tables 3 and 4 for the ORB maximum and the inner edge, respectively. The linear fits are compared with geomagnetic field trends predicted by the IGRF model. The trends are also fitted by a linear function with the slope $a_{IGRF}$."

2. The errors of the fitting parameters have been calculated by using a linear regression. The results are listed in new tables (Tables 3 and 4) and discussed in the text.

3. The fitting expressions have been removed from Figures 4 and 5.

4. Description of Figures 4 and 5 was substantially improved with using discussion of the linear regression results presented in Tables 3 and 4.

5. The comparison with the IGRF model in the Siberian sector was revised in the following manner:

"From Table 4, one can see that the slopes $a$ are steeper than $a_{IGRF}$. The slopes are calculated with errors of ~30% and ~20%, respectively, for >30 keV and >100 keV electrons. It means that the decrease in latitude might be ~2° (instead of ~3°) and ~1.5° (instead of ~2°), respectively. These values are again larger than 1° of the model prediction. Hence, there is a tendency that the change in the latitudinal location of ORB maximum is underestimated by the model. This fact indicates that during 17 years from 2001 to 2018, ORB is abnormally displaced toward the lower latitudes in the Siberian sector."

6. The abstract and conclusions have been revised accordingly:

"However in the Siberian sector, the model has a tendency to underestimate the equatorward shift of ORB."

**Reply to Reviewer's #2 comments.**
I am grateful to the reviewer for very useful and valuable comments and suggestions. They help to improve substantially the quality of manuscript.

The following revisions have been done in according to the comments (colored by blue in the text):

Comment 1
*In lines 27 – 28, a brief description of the outer Van Allen radiation belt is provided where this population of charged particles is presented as part of the outer magnetosphere, contrary to what has been widely established and is presented in the following publications:*
*- Baker (1995), The inner magnetosphere: A review, Surveys in Geophysics, doi: 10.1007/BF01044572*
*- Ebihara & Miyoshi (2011), Dynamic inner magnetosphere: A tutorial and recent advances,*
*in Liu W., Fujimoto M. (eds) The dynamic magnetosphere, doi: 10.1007/978-94-007-0501-2_9*

Reply 1
The sentences were revised accordingly:
"The outer radiation belt (ORB) is populated by energetic and relativistic electrons trapped in the magnetosphere at drift shells above $L \sim 3$ (e.g. Ebihara and Miyoshi, 2011). The ORB is very dynamic and exhibits variations…"
The term "outer magnetosphere" has been removed from whole the text.

Comment 2
*Additional references on the long-term variations of the radiation belts' structure that should be considered are the following:*
*- Fung et al. (2006), Long-term variations of the electron slot region and global radiation belt structure, Geophysical Research Letters, doi: 10.1029/2005GL024891*
*- Baker & Kanekal (2008), Solar cycle changes, geomagnetic variations, and energetic particle properties in the inner magnetosphere, Journal of Atmospheric and Solar-Terrestrial Physics, doi: 10.1016/j.jastp.2007.08.031*
*- Glauert et al. (2018), A 30-year simulation of the outer electron radiation belt, Space Weather, doi: 10.1029/2018SW001981*

*On lines 30 and 35, the semi-annual variation of the outer radiation belt is mentioned. This seasonal and not annual change could be explained by the IMF-effect also known as Russell-McPherron effect which is described in:*
*- Russell & McPherron (1973), Semiannual variation of geomagnetic activity, Journal of Geophysical Research, doi: 10.1029/JA078i001p00092*
*- McPherron et al. (2009), Role of the Russell-McPherron effect in the acceleration of relativistic electrons, Journal of Atmospheric and Solar-Terrestrial Physics, doi: 10.1016/j.jastp.2008.11.002*
*and where origins of the seasonal variability in geomagnetic activity have been traced.*

Reply 2
The first paragraph of Introduction section has been revised accordingly:

"The ORB is very dynamic and exhibits variations in a wide temporal range: short-term storm-time and local time variations, 27-day solar rotation, seasonal and solar cycle variations (e.g. Li et al., 2001; Baker and Kanekal, 2008; Miyoshi and Kataoka, 2011). During magnetic storms, the ORB is substantially disturbed and shifted earthward (Baker et al., 2016; Shen et al., 2017). The storm-time variation is the strongest one for both the ORB location and intensity (Baker and Kanekal, 2008). Magnetic storms produced by interplanetary coronal mass ejecta (ICME) and high-speed streams (HSS) of the solar wind from coronal holes. The seasonal variations with maxima at equinoxes can be explained by the effect of interplanetary magnetic field (IMF) orientation relative to the geomagnetic dipole (Li et al., 2001; McPherron et al., 2009). ORB manifests prominent variations with the solar cycle (Fung et al., 2006; Baker and Kanekal, 2008). It was shown that the maximum of ORB is mostly distant from the Earth in solar minimum (Miyoshi et al., 2004) and it is closest to the Earth during solar maxima (Glauert et al., 2018)."

Comment 3
*There are minor issues with English language use and several typographical errors.*
*For example, on line 54, the term auroral electrojet is first introduced that should be one word. Since Smith et al. (2017) studied both current in the north and south hemisphere, it should also be plural (auroral electrojets – AEJs).*

Reply 3
Corrected

Comment 4
*Further down on the same page, on line 72, the work of Kataoka et al. (2015) is listed among the references for the October-November 2003 superstorms although it is focused on the magnetic storm of 17 March 2015 which is mentioned further down, in lines 74-81. It should, therefore, be moved more appropriately further down after the brief description of the 2015 Saint Patrick's Day storm.*

Reply 4
Thank you for suggestion. Corrected.

Comment 5
*On line 89, it is indicated that the satellite observations used for the study of the outer electron belt location cover the time period from 1998 to 2016. However, the dates listed in Table 2 are within the time period between 2001 and 2018 which is also the time period on which this study is focused as indicated throughout the manuscript.*

Reply 5
The year 1998 is replaced with 2001

Comment 6
*On line 119, it should read "geomagnetic activity was very weak" and not "very week".*

Reply 6
Corrected

Comment 7
*On line 135, it is indicated that Figure 2 shows POES observations from 3 June 2016, while the figure caption indicated that the observations shown are from 2 June 2016.*

Reply 7
Yes, it was misprinting. In Figure 2 caption, 2 June was replaced with 3 June.

Comment 8
*In lines 123-126, the close link between increases in solar wind speed and enhancements in electron fluxes in the outer radiation belt is briefly described. Periodic oscillations in the Earth's magnetic field with frequencies in the range of a few millihertz (ultralow frequency waves) may indeed be an intermediary through which solar wind influences radiation belt dynamics due to their potential for resonant interactions with energetic electrons causing the radial migration of resonant electrons. It should, however, be corrected that electrons are accelerated and increase their energy when they are transported earthward to regions of stronger geomagnetic field. Recent, representative publications on this acceleration mechanism are the following:*
*- Mann et al. (2013), Discovery of the action of a geophysical synchrotron in the Earth's Van Allen radiation belts, Nature Communications, doi: 10.1038/ncomms3795*
*- Su et al. (2015), Ultra-low frequency wave-driven diffusion of radiation belt relativistic electrons, Nature Communications, doi: 10.1038/ncomms10096*
*Radial transport acts as a loss mechanism when particle drift outward and are lost to the magnetopause. The work of Horne et al. (2007) and Reeves et al. (2013), provided as reference, is centered on a different acceleration mechanism acting in the heart of the radiation belt (local acceleration) that involves whistler mode chorus waves rather than waves generated through the Kelvin-Helmholtz instability along the magnetopause.*

Reply 8
Thank you for very useful papers. Note that storm-time acceleration and transport of energetic electrons in ORB is not the subject of the present study, which is dealing with quiet days. Hence, this part has been revised accordingly:
"Note that the solar wind with the speed of V > 400 km/s is often associated with HSSs from coronal holes. Fast solar wind streams initiate the Kelvin-Helmholtz instability at the magnetopause and also produce recurrent magnetic storms, which are accompanied by intensification of wave activity in the outer magnetosphere that results in effective acceleration and radial transport of the ORB electrons (Engebretsone et al., 1998; Tsurutani et al., 2006; Horne et al., 2007; Su et al., 2015)."

Comment 9
*In lines 187-188, the author notes that the inner edge of the outer radiation is defined as the "first high-latitude point of electron flux enhancements". Could the latitude above which the flux enhancement was*

*searched be indicated? In addition, which criterion was applied on flux measurements to determine which fluctuations in electron flux correspond to the enhancement observed at the inner edge of the outer electron belt?*

Reply 9
This important issue is described in more details in the revised manuscript:
"In Figure 3, one can clearly see the slot region between the outer and inner electron belts in the latitudinal ranges 45° - 50° and 45° - 50° above North America and Siberia, respectively. This structure can be well identified and numerically determined, excepting >300 keV electrons. In the case of slot region, the low-latitude edge of ORB is determined as the first high-latitude point of gradual flux enhancement after the slot minimum. Apparently, the electron flux enhancements peak in the maximum of ORB, which location can be determined unambiguously. In the European sector and for the electrons with energies >300 keV, the criterion for determination of the inner edge is not so obvious. It is difficult to define a threshold flux because of strong solar cycle variations of electron fluxes. In this case, the inner edge can be determined as the lowest latitude of gradual decrease of electron fluxes from the ORB maximum toward lower latitudes. As one can see in Figure 3, the inner edge separates usually the background noise with sharply varying fluxes at lower latitudes from smooth and fast increase of ORB fluxes at higher latitudes. Geographic latitude of the inner edge is determined for each year with the accuracy varying from 0.5° to 1°. In the American sector, the inner edge of ORB is situated at lowest latitudes from 43° to 51°, in the European sector – from 55° to 63°, and in the Siberian sector – at highest latitudes from 58° to 65°."

Comment 10
*On line 203, the maximum of solar cycle 23 is indicated that it was observed in 2001 and that of solar cycle 24 in 2012-2013. It is not clear to me, and perhaps the reader, how this maximum was defined as both solar cycles were double-peaked according to the number of sunspots observed on the surface of the Sun that has been presented in Figures 4 and 5 with the solid grey line.*

Reply 10
Solar maximum, as a physical phenomenon of the solar magnetic field reversal, has the double-peak structure both in the $23^{rd}$ and $24^{th}$ solar cycles. Those cycles peaked, respectively, in November 2001 and in April 2014. After the peaks, the declining phases started and the solar activity was quickly decreasing. Dramatic changes in the solar magnetic field (including the reversals) were observed from 2000 to 2001 and from 2012 to the beginning of 2014, respectively. In this sense, the maximal phases of those cycles occurred definitely in 2001 - 2002 and 2012 - 2013. The year of 2014 belongs both to the maximum and to the declining phase of the $24^{th}$ solar cycle.
In the original paper, the year 2000 was not shown and, hence, it was mentioned.
In the revised manuscript, this issue is described in more details:
"Note that the maximum phases of the $23^{rd}$ and $24^{th}$ solar cycles occurred in the years 2000 - 2001 and in 2012 – April 2014, respectively. The years 2008 – 2009 are the solar minimum phase. The declining phases lasted from 2003 to 2007 and from 2014 to 2018. In Figures 4 and 5, one can see that during the declining phase of the current $24^{th}$ solar cycle (especially in the years 2016 – 2018), the behavior of the ORB maximum and inner edge is different from that during the declining phase of the previous $23^{rd}$ solar cycle. Namely, their latitudes increased only slightly or even decreased above North America and especially above Siberia."

Comment 11
*Further down, in the paragraph starting with line 208, an example of how the IGRF-12 model was used on the geographic coordinates of the outer radiation belt maximum flux to obtain the corresponding geomagnetic coordinates is provided. What height was selected as input to the model to determine the geomagnetic or geographic coordinates?*

Reply 11
The height of POES orbit at 850 km was used. Actually for a given magnetic coordinates, the long-term changes in geographic latitudes predicted by the IGRF-12 model do not vary much with the heights for the low-earth orbits and ground (10% of the Earth's radius). The latitudes are different, but their changes with time are almost same.

Comment 12
*The choice of a simple linear fit over the set of outer radiation belt latitudes that have been calculated for the selected quiet days in the period 2001-2018 and are presented in Figures 4 and 5 puzzles me as it seems inadequate to support the main conclusion of the study. There is significant variability in the outer radiation belt location that is related to the solar activity variability that has not been accounted for – although it should - during linear regression.*

Reply 12
The using of linear fit can be justified in the following way:
"Unfortunately, there is no any model of the ORB location variation with the solar cycle because the driving mechanisms are not well established. On the other hand, the long-term variation in IGRF-12 is almost linear function of the year, as one can see in Figures 4 and 5. Hence, as a first approach for comparative analysis, the variations of ORB location with years are considered as random around a linear function (indicated by dashed strait lines in Figures 4 and 5):

$$\lambda = a * \text{year} + b, (1)$$

where $\lambda$ is the latitude of maximum or inner edge of ORB. The slope $a$, parameter $b$ and their standard errors are calculated from a linear regression for various longitudinal regions and various energies of electrons. The results are presented in Tables 3 and 4 for the ORB maximum and the inner edge, respectively. The linear fits are compared with geomagnetic field trends predicted by the IGRF model. The trends are also fitted by a linear function with the slope $a_{IGRF}$."

Furthermore, the solar cycle variations of ORB are investigated in Discussion section of revised manuscript with using a new Figure 6.
"In Figure 6, the outer radiation belt location is compared during the maximum and declining phases of the solar cycles 23rd (the years 2001 – 2006) and 24th (the years 2013 – 2018). During those time intervals, the sunspot numbers for the both cycles correlate very well. The ORB location demonstrates also very similar solar cycle variations. The ORB latitude increased after the solar maximum in 2001 –

2002 (and in corresponding years 2013 – 2014). During those years, the ORB location was quite close for the both cycles. The difference of ~1° can be explained by the secular variation predicted by the IGRF model. During the declining phase in 2004 – 2005 (2015 – 2017), the ORB was shifted to lower latitudes and then it moved slightly poleward in 2006 (2018), when the solar minimum was approached.

From Figure 6, one can clearly see that on the declining phase of the 24[th] solar cycle, the outer radiation belt is located at latitudes lower by several decrees than those during the 23[rd] solar cycle. It is interesting to point out the year 2017, when the maximum and inner edge of ORB were shifted to very low latitudes of 62° and ~59° respectively. The shift was observed during two quiet days on 9 and 10 June 2017. Similar pattern of strong displacement by more than ~2° can be found on the declining phase of the previous 23rd solar cycle in 2005, the year corresponding to the similar stage of solar activity. Hence, the ORB dynamics in the year of 2017, as well as during the whole declining phase from 2014 to 2018, was not anomalous in the sense of solar cycle variations. However, the ORB latitudes were abnormally low. The difference of several degrees cannot be explained by the IGRF model. As a result, we have found totally opposite effect: ORB over Siberia is located at lower latitudes during the weak 24th solar cycle than during the strong 23rd solar cycle. It should be noted that if one excludes the year 2017 from the linear fitting then the results are not practically changed because ORB is located at relatively low latitudes during practically whole declining phase of the 24th solar cycle. ”

[Figure]

Figure 6. Geographic latitude of the inner edge (a) and maximum (b) of the outer radiation belt measured during geomagnetic quiet days at height of ~850 km around longitude of 100°E for electrons with energies of >30 keV (circles), >100 keV (crosses), and >300 keV (triangles). Bottom panels show the sunspot number in the 23ʳᵈ solar cycle (the years 2001 – 2006, black curves) and in the 24ᵗʰ solar cycle (the years 2013 – 2018, blue curves). The outer radiation belt location is shown by black and red symbols, respectively, for the 23ʳᵈ and 24ᵗʰ solar cycles. It can be seen that on the declining phase of the 24ᵗʰ solar cycle, the outer radiation belt is systematically located at lower latitudes than that during the 23ʳᵈ solar cycle.

Comment 13
*The difference in the variability observed in the location of the inner edge of the outer radiation belt or the location of the maximum electron flux has also not been quantified nor considered in the evaluation of the difference estimated between electron observations and magnetic field predictions from the IGRF-12.*

Reply 13
An effort of inter comparison between the ORB inner edge and maximum was made in the original paper:
"As can be seen in Figures 4 and 5, the location of ORB manifests the well-known solar cycle variation: the latitudes of ORB maximum and inner edge have a tendency to be highest around solar minimum in 2008 – 2009 and lowest during solar maxima in the years 2001 and 2012 – 2013."
In the revised manuscript, the difference is quantified and discussed for each longitudinal sector. For example:
"Similar pattern can be found for the inner edge of ORB in the Siberian sector (see Figure 5c). Namely, the IGRF model predicts a decrease of ~1° with the slope $a_{IGRF}$ ~ -0.06. The inner edge was shifted toward lower latitudes by ~3°, ~2° and ~1°, respectively, for >30 keV, >100 keV and >300 keV electrons. From

Table 4, one can see that the slopes *a* are steeper than $a_{\text{IGRF}}$. The slopes are calculated with errors of ~30% and ~20%, respectively, for >30 keV and >100 keV electrons. It means that the decrease in latitude might be ~2° (instead of ~3°) and ~1.5° (instead of ~2°), respectively. These values are again larger than 1° of the model prediction. Hence, there is a tendency that the change in the latitudinal location of ORB maximum is underestimated by the model. This fact indicates that during 17 years from 2001 to 2018, ORB is abnormally displaced toward the lower latitudes in the Siberian sector."

Comment 14
*On line 264, among the reference provided for the effect of the tilt angle variation on the location of the outer radiation belt, the study of Newell et al. (2006) is found. The specific study was centred on the cusp location as it is detailed in the next paragraph and should, therefore, be excluded from the reference list provided here.*

Reply 14
Corrected, the reference has been replaced.

Comment 15
*In lines 269 and 270, the cusp location is suggested as a proxy of the outer radiation boundary. The author must imply the outer edge of the electron radiation belt or more correctly that a displacement of the cusp influences the location of the outer radiation belt but this is not clear from the text. It is also not substantiated by the findings of Newell et al. (2006).*
*To date, the inner edge of the outer radiation belt has been suggested to be defined by the plasmapause, the outer boundary of the plasmasphere. Specifically, in the following publication:*
*- Baker et al. (2014), Impenetrable barrier to ultrarelativistic electrons in the Van Allen radiation belts, Nature, doi: 10.1038/nature13956 the authors analysed 20 months of electron flux data from the NASA/Van Allen Probes to identify a barrier in the inward transport of ultrarelativistic electron transport. Earlier, - Darrouzet et al. (2013), Links between the plasmapause and the radiation belt boundaries as observed by the instruments CIS, RAPID and WHISPER onboard Cluster, Journal of Geophysical Research, doi: 10.1002/jgra.50239*
*had reached a different conclusion. The radiation belt location was found to be dependent on the energy range of particles examined but also that the plasmapause is more variable that the inner edge of the outer radiation belt. Namely, the inner or outer edge of the outer electron belt does not always coincide with the plasmapause.*

Reply 15
Thank you very much for very useful papers.
This paragraph has been revised accordingly:
"The effect of solar wind parameters, including IMF *B*z and dynamic pressure (*P*d), to the ORB location is not obvious. It is found that the slot region location can be related to the plasmapause but the relation is ambiguous (Darrouzet et al., 2013; Baker et al., 2014). We can make indirect estimation of the effect using a dependence of the cusp location from the solar wind parameters (Kuznetsov et al., 1993; Newell et al., 2006). The equatorward edge of the cusp separates the open and close magnetic filed lines in the dayside magnetosphere. Hence the latitude of the equatorward edge can be considered as a proxy of the ORB outer edge. In the first approach, we assume that the effect of solar wind parameters to the ORB

location can be represented by the dynamics of the ORB outer edge or the cusp equatorward edge. It can be shown that $Bz$ = -4 nT results in less than 0.5° equatorward shift of the cusp and a change of $Pd$ from 1 to 2 nPa results in ~0.2° decrease in the latitude of the cusp equatorward edge. Hence, the effects of both $Pd$ and IMF Bz are several times weaker than the difference of 3°."

Comment 16
*In lines 274 and 275, the statement "Variations of the ORB location from cycle to cycle are not investigated yet" is not entirely correct. There are indeed significant limitations in such studies due to the lack of data covering several years that could be discussed at this point. Reference to studies such as Glauert et al. (2018) could also be provided.*

Reply 16
Thank you very much for very useful paper. This part was revised accordingly:
"Variations of the ORB location from cycle to cycle and during different phases of solar cycles are still poorly investigated. It was well established that during solar minima and maxima, the ORB is located, respectively, at highest and lowest latitudes (Miyoshi et al., 2004; Glauert et al., 2018). From these findings, we can speculate that lower(higher) solar activity results in an increase (a decrease) of the ORB latitudes."

Comment 17
*On line 295, the findings of Finlay et al. (2015) suggesting rapid changes in the geomagnetic field in the past 15 years are briefly mentioned. Although the latest change observed in 2012-2013 seems to influence the location of the outer radiation belt, is there a signature of the change observed in 2006 and 2009 in the POES measurements from the same period analysed here?*

Reply 17
This important issue is clarified in the following manner:
"We can assume that the abnormal ORB displacement might be related to the geomagnetic jerks. Though, there is no prominent change in the ORB location in 2006, one can indicate very high latitude of ORB in 2009. Note that the jerk in 2009 coincided with the abnormally deep solar minimum and, hence, it could be hard to distinguish between the two effects. On the other hand, we have found significant change in the ORB dynamics after 2012 – 2013."

Comment 18
*Individual graphs in Figure 2 are difficult to read because of the dark background colour.*
*The font size selected for the x and y axis labels are so small that, even after blowing them up to 200%, labels are still difficult to read. On the other hand, titles over the two columns (2016 for the right column and 2006 for the left column) seem to be misplaced.*

Reply 18
Figure 2 has been revised accordingly.

Comment 19
*Fonts on plots in Figure 3 could also be enlarged if these are selected to be the final sizes of the graphs.*

Reply 19
The multi-plot Figure 3 is mainly presented in order to demonstrate qualitatively the structure of ORB and its dynamics in various longitudinal sectors. The results of numerical analysis are presented in Figures 4 and 5.

**Reply to Editor's comments**
Dear Elias Roussos,

I appreciate your helpful comments and suggestions. The manuscript was further revised accordingly. The revisions are colored by blue in the text.

Sincerely yours,
Alexei Dmitriev

*2) On the subject of the definition of ORB: one of the reviewers asked how is this obtained. You clarified that you define it as a starting point that a continuous increase is seen in the fluxes. However, I may have missed if you also require that a certain flux limit or S/N has to be be crossed. If so, please elaborate further. E.g. a continuous increase of fluxes can be slow or sharp - more impulsive. Do you define the boundary for such occasions in a different way?*

The description of the criteria for determination of the ORB inner edge and maximum has been revised:
"In Figure 3, one can clearly see the slot region between the outer and inner electron belts in the latitudinal ranges 45° - 50° and 45° - 50° above North America and Siberia, respectively. This structure can be well identified and numerically determined, excepting >300 keV electrons. In the case of slot region, the low-latitude edge of ORB is determined as the first high-latitude point of gradual flux enhancement after the slot minimum. Apparently, the electron flux enhancements peak in the maximum of ORB, which location can be determined unambiguously. In the European sector and for the electrons with energies >300 keV, the criterion for determination of the inner edge is not so obvious. It is difficult to define a threshold flux because of strong solar cycle variations of electron fluxes. In this case, the inner edge can be determined as the lowest latitude of gradual decrease of electron fluxes from the ORB maximum toward lower latitudes. As one can see in Figure 3, the inner edge separates usually the background noise with sharply varying fluxes at lower latitudes from smooth and fast increase of ORB fluxes at higher latitudes."

*1) On the subject of linear fitting of the ORB location and the conclusion about its large, 2-3 deg latitudinal "offset". While I understand that variations seem to a first order random and a linear fitting is a good first approximation, the errors of the linear fitting parameters are comparable to the derived, averaged ORB displacements, even though for certain longitudes absolute values of errors are smaller. Inspection of the corresponding Fig. 4 and 5 plots give the impression that the linear fitting parameters could be misleading. E.g., it should be investigated how fittings are affected by outlier points: in Fig.4 there are clear, extreme low ORB values in 2017 (top panel) and 2012 (bottom panel). As the are potential outliers which may control the fitting and impose a negative latitudinal gradient, their origin could be discussed. Also, a few deg change in ORB over more than a solar cycle should be obtained even if you e.g. remove 1-1.5 years of data either from the start or the end of the data series - that could be a further test of your findings. It was also not clear to me if error propagation was used in the linear fitting (i.e. do you take into account the uncertainty in each data point?).*

The error propagation is not used in the linear fitting because the data errors are similar (0.5 and 1 deg).

In order to consider the origin and possible effects of "extremely low" values, an additional comparative analysis was performed for the solar cycle variation of the ORB location and an additional Figure 6 was introduced. This part was included in Discussion Section:

"…Following this logic, the ORB should be located at relatively higher latitudes during the weak 24th solar cycle than during the strong 23rd solar cycle.

In Figure 6, the outer radiation belt location is compared during the maximum and declining phases of the solar cycles 23rd (the years 2001 – 2006) and 24th (the years 2013 – 2018). During those time intervals, the sunspot numbers for the both cycles correlate very well. The ORB location demonstrates also very similar solar cycle variations. The ORB latitude increased after the solar maximum in 2001 – 2002 (and in corresponding years 2013 – 2014). During those years, the ORB location was quite close for the both cycles. The difference of ~1° can be explained by the secular variation predicted by the IGRF model. During the declining phase in 2004 – 2005 (2015 – 2017), the ORB was shifted to lower latitudes and then it moved slightly poleward in 2006 (2018), when the solar minimum was approached.

From Figure 6, one can clearly see that on the declining phase of the 24[th] solar cycle, the outer radiation belt is located at latitudes lower by several decrees than those during the 23[rd] solar cycle. It is interesting to point out the year 2017, when the maximum and inner edge of ORB were shifted to very low latitudes of 62° and ~59° respectively. The shift was observed during two quiet days on 9 and 10 June 2017. Similar pattern of strong displacement by more than ~2° can be found on the declining phase of the previous 23rd solar cycle in 2005, the year corresponding to the similar stage of solar activity. Hence, the ORB dynamics in the year of 2017, as well as during the whole declining phase from 2014 to 2018, was not anomalous in the sense of solar cycle variations. However, the ORB latitudes were abnormally low. The difference of several degrees cannot be explained by the IGRF model. As a result, we have found totally opposite effect: ORB over Siberia is located at lower latitudes during the weak 24th solar cycle than during the strong 23rd solar cycle. It should be noted that if one excludes the year 2017 from the linear fitting then the results are not practically changed because ORB is located at relatively low latitudes during practically whole declining phase of the 24th solar cycle."

[Figure]

Figure 6. Geographic latitude of the inner edge (a) and maximum (b) of the outer radiation belt measured during geomagnetic quiet days at height of ~850 km around longitude of 100°E for electrons with energies of >30 keV (circles), >100 keV (crosses), and >300 keV (triangles). Bottom panels show the sunspot number in the 23$^{rd}$ solar cycle (the years 2001 – 2006, black curves) and in the 24$^{th}$ solar cycle (the years 2013 – 2018, blue curves). The outer radiation belt location is shown by black and red symbols, respectively, for the 23$^{rd}$ and 24$^{th}$ solar cycles. It can be seen that on the declining phase of the 24$^{th}$ solar cycle, the outer radiation belt is systematically located at lower latitudes than that during the 23$^{rd}$ solar cycle.